# miR-181a/b downregulation exerts a protective action on mitochondrial disease models

Alessia Indrieri[1,2,‡] (ID), Sabrina Carrella[1,3,‡], Alessia Romano[1], Alessandra Spaziano[1], Elena Marrocco[1], Erika Fernandez-Vizarra[4] (ID), Sara Barbato[1], Mariateresa Pizzo[1], Yulia Ezhova[1], Francesca M Golia[1], Ludovica Ciampi[1], Roberta Tammaro[1], Jorge Henao-Mejia[5,6], Adam Williams[7,8], Richard A Flavell[9,10], Elvira De Leonibus[1,11], Massimo Zeviani[4] (ID), Enrico M Surace[1,2,†], Sandro Banfi[1,3,*] (ID) & Brunella Franco[1,2,**] (ID)

## Abstract

Mitochondrial diseases (MDs) are a heterogeneous group of devastating and often fatal disorders due to defective oxidative phosphorylation. Despite the recent advances in mitochondrial medicine, effective therapies are still not available for these conditions. Here, we demonstrate that the microRNAs miR-181a and miR-181b (miR-181a/b) regulate key genes involved in mitochondrial biogenesis and function and that downregulation of these miRNAs enhances mitochondrial turnover in the retina through the coordinated activation of mitochondrial biogenesis and mitophagy. We thus tested the effect of miR-181a/b inactivation in different animal models of MDs, such as microphthalmia with linear skin lesions and Leber's hereditary optic neuropathy. We found that miR-181a/b downregulation strongly protects retinal neurons from cell death and significantly ameliorates the disease phenotype in all tested models. Altogether, our results demonstrate that miR-181a/b regulate mitochondrial homeostasis and that these miRNAs may be effective gene-independent therapeutic targets for MDs characterized by neuronal degeneration.

**Keywords** LHON; microRNA; miR-181; mitochondrial disease; neurodegeneration

**Subject Categories** Genetics, Gene Therapy & Genetic Disease; Pharmacology & Drug Discovery

## Introduction

Mitochondrial diseases (MDs) represent a relevant group of inherited disorders with a cumulative prevalence of about 1:5,000 individuals (Gorman *et al*, 2015). They are caused by mutations in either nuclear or mitochondrial genes resulting in oxidative phosphorylation (OXPHOS) impairment, leading to huge variability of symptoms, organ involvement, and clinical course. The clinical manifestations range from dysfunction of single tissue/structures such as the optic nerve in Leber's hereditary optic neuropathy (LHON, MIM535000), to syndromic multi-organ conditions with a prominent involvement of the central nervous system (CNS), such as microphthalmia with linear skin lesions (MLS, MIM309801, 300887, 300952) and Leigh syndrome (LS, MIM256000). Neurons are particularly sensitive to mitochondrial dysfunction due to their highest energy demands, and defects in mitochondrial metabolism may lead to severe energy deficiency, increased reactive oxygen species (ROS), and neuronal death.

MDs are biochemically and genetically heterogeneous, and their complexity has so far prevented the development of effective treatments (Sanchez *et al*, 2016). In the past few years, specific

1   Telethon Institute of Genetics and Medicine (TIGEM), Pozzuoli, Italy
2   Medical Genetics, Department of Translational Medical Science, University of Naples "Federico II", Naples, Italy
3   Medical Genetics, Department of Precision Medicine, University of Campania "L. Vanvitelli", Caserta CE, Italy
4   MRC Mitochondrial Biology Unit, University of Cambridge, Cambridge, UK
5   Department of Pathology and Laboratory Medicine, University of Pennsylvania, Philadelphia, PA, USA
6   Institute for Immunology, Perelman School of Medicine, University of Pennsylvania, Philadelphia, PA, USA
7   The Jackson Laboratory for Genomic Medicine, Farmington, CT, USA
8   Department of Genetics and Genomic Sciences, University of Connecticut Health Center, Farmington, CT, USA
9   Department of Immunobiology, Yale University School of Medicine, New Haven, CT, USA
10  Howard Hughes Medical Institute, Chevy Chase, MD, USA
11  Institute of Cellular Biology and Neurobiology "ABT", CNR, Roma, Italy
    *Corresponding author. Tel: +39 08119230606; E-mail: banfi@tigem.it
    **Corresponding author. Tel: +39 08119230605; E-mail: franco@tigem.it
    ‡These authors contributed equally to this work
    †Present address: Medical Genetics, Department of Translational Medical Science, University of Naples "Federico II", Naples, Italy

modulation of either mitochondrial biogenesis/dynamics or mitochondrial clearance/quality control has been tested as possible therapeutic strategies in different MD models (Viscomi *et al*, 2011; Johnson *et al*, 2013; Cerutti *et al*, 2014; Civiletto *et al*, 2015, 2018). In spite of initial promising data, the latter approaches failed to be effective across different MD models [reviewed in Lightowlers *et al* (2015); Viscomi *et al* (2015)]. We hypothesize that a synergic and fine modulation of mitochondrial biogenesis and clearance pathways is necessary to ensure a more efficient neuroprotective effect.

MicroRNAs (miRNAs) are fundamental fine regulators of gene expression and represent promising therapeutic tools due to their capability of simultaneously modulating multiple molecular pathways involved in disease pathogenesis and progression. Modulation of miRNAs has been applied, with therapeutic purposes, to different disorders and has reached preclinical and clinical stages in specific instances (Broderick & Zamore, 2011; Janssen *et al*, 2013; Ling *et al*, 2013; Christopher *et al*, 2016). miRNAs play a key role in neuron survival, and accumulating evidence indicates that alterations of miRNA-regulated gene networks increase the risk of neurodegenerative disorders (Hebert & De Strooper, 2009). miR-181a and miR-181b (miR-181a/b) belong to a family of miRNAs highly expressed in different regions of brain and retina (Boudreau *et al*, 2014; Karali *et al*, 2016) and were recently reported to target genes involved in mitochondrial-dependent cell death (Ouyang *et al*, 2012; Hutchison *et al*, 2013; Rodriguez-Ortiz *et al*, 2014) and autophagy (He *et al*, 2013; Tekirdag *et al*, 2013; Cheng *et al*, 2016).

Here, we show that miR-181a/b are involved in global regulation of mitochondrial function by controlling a group of genes involved in mitochondrial biogenesis and function, and redox balance. We demonstrate that downregulation of these two miRNAs protects retinal neurons from mitochondrial dysfunction, and ameliorates the phenotype of three different MD animal models with ocular involvement, indicating that miR-181a/b could be new therapeutic targets for MDs.

# Results

## miR-181a/b control mitochondrial turnover

Bioinformatic search (Gennarino *et al*, 2012) allowed us to identify *PPARGC1A* and *NRF1*, master regulators of mitochondrial biogenesis (Wu *et al*, 1999; Finck & Kelly, 2006), *COX11* and *COQ10B*, involved in mitochondrial respiratory chain (MRC) assembly (Carr *et al*, 2002; Desbats *et al*, 2015), and *PRDX3*, an important mitochondrial ROS scavenger (Wonsey *et al*, 2002), as putative miR-181a/b target genes. Interestingly, quantitative real-time PCR (qPCR) analysis indicated increased levels of all of the above-mentioned transcripts in SH-SY5Y human neuroblastoma cells following miR-181a/b silencing (Fig 1A). To validate the newly predicted miR-181a/b targets, 3′-UTRs of each human gene (*PPARGC1A, NRF1, COX11, COQ10B,* and *PRDX3*), including the predicted miR-181 target site, were cloned in the pGL3-TK-luciferase plasmid, downstream the coding region of the luciferase reporter gene. We then tested the ability of transfected mimic-miR-181 to affect luciferase activity. The presence of the 3′-UTR sequence of the analyzed genes inhibited luciferase activity in response to mimic-miR-181 (Fig 1B). In addition, point mutations in the miR-181a/b binding site in the 3′-UTR

of each gene abolished luciferase repression, demonstrating that these miRNAs directly and specifically target *NRF1, COX11, COQ10B,* and *PRDX3* (Fig 1B). Direct targeting of *PPARGC1A* was not validated (Fig 1B), indicating that the upregulation observed by qPCR after miR-181a/b silencing (Fig 1A) could be the result of an indirect effect.

Based on the above results, we reasoned that the downregulation of miR-181a/b could stimulate mitochondrial biogenesis and decided to test this hypothesis *in vivo*. In mammals, miR-181a and miR-181b are organized in two clusters, namely *miR-181a/b-1* and *miR-181a/b-2*, which are localized to different genomic loci. The mature forms of miR-181a-1 and miR-181a-2, as well as those of miR-181b-1 and miR-181b-2, display identical sequences. Furthermore, both miR-181a and miR-181b contain the same "seed" sequence (Ji *et al*, 2009), i.e., the region that is believed to play the most important role in target recognition (Bartel, 2009). We chose to analyze a mouse model harboring a targeted deletion of the *miR-181a/b-1* cluster (Henao-Mejia *et al*, 2013). This cluster accounts for most of the expression of mature miR-181a/b in the retina, as demonstrated by RNA *in situ* hybridization, TaqMan assays, and the increase in several previously validated miR-181a/b targets, such as *Bcl2, Mcl1, Atg5, Erk2,* and *Park2* (Ouyang *et al*, 2012; He *et al*, 2013; Hutchison *et al*, 2013; Tekirdag *et al*, 2013; Rodriguez-Ortiz *et al*, 2014; Cheng *et al*, 2016) (Appendix Fig S1, Fig EV1A).

Interestingly, by qPCR, we observed increased expression levels of *Nrf1, Cox11, Coq10b Prdx3,* and *Ppargc1a* in the eye of *miR-181a/b-1$^{-/-}$* mice (Fig 1C). Upregulation of *Nrf1* and *Ppargc1a* indicates enhanced mitochondrial biogenesis (Wu *et al*, 1999; Finck & Kelly, 2006). In line with this observation, we detected an increase of mitochondrial DNA (mtDNA), as measured by qPCR (Fig 1D), and of the protein levels of MRC complex subunits (Ndufb11 and CoxIV), mitochondrial matrix (citrate synthase [Cs]), and mitochondrial membranes (Mfn2 and Tim23), as assessed by Western blot (WB) analysis in the eye of *miR-181a/b-1$^{-/-}$* mice (Fig 1E). Overall, these data demonstrate that miR-181a/b inactivation stimulates mitochondrial biogenesis in the CNS.

It was previously reported that miR-181a/b regulate *in vitro* the expression of *Atg5* and *Park2*, which are key players in autophagy and mitophagy (Tekirdag *et al*, 2013; Cheng *et al*, 2016). Therefore, we investigated whether miR-181a/b inactivation enhances mitophagy in the mouse eye. First, we observed, by qPCR assays, that the transcript levels of the *Atg5* and *Park2* genes were upregulated in *miR-181a/b-1$^{-/-}$* eyes (Fig EV1A). Moreover, we demonstrated, by WB, enhanced recruitment of *Park2*/Parkin and of the autophagic adaptor *Sqstm1*/p62 in mitochondrial fractions (Fig 1F), indicating an increase of mitophagy in *miR-181a/b-1$^{-/-}$* eyes. We also analyzed general autophagy by evaluating the levels of *Sqstm1*/p62 and of the autophagic marker *Map1lc3b*/Lc3-II in total protein extracts. By WB, we observed decreased levels of these two proteins, which is more evident in starved conditions (Fig EV1B), without changes in the corresponding transcript levels (Fig EV1C), indicating an increase in the autophagic flux rate in *miR-181a/b-1$^{-/-}$* eyes.

Taken together, these results uncover an important role of miR-181a/b in the regulation of mitochondrial turnover through the coordination of mitochondrial biogenesis and clearance *in vivo*.

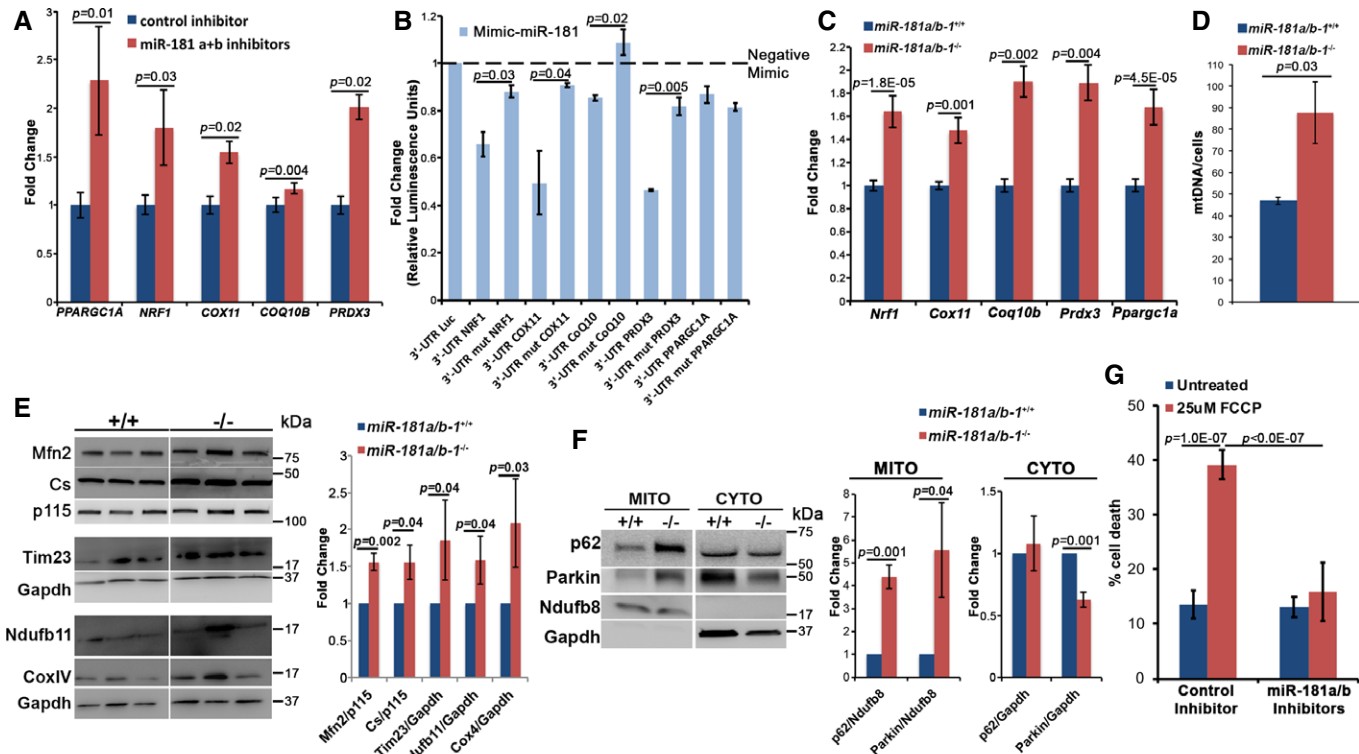

**Figure 1. miR-181a/b silencing increases mitochondrial biogenesis and mitophagy and protects retinal neurons from FCCP-induced cell death.**

A  qPCR reveals that miR-181a/b silencing leads to upregulation of miR-181a/b predicted targets in SH-SY5Y cells. $N = 3$ independent experiments.

B  miR-181-mimic transfection specifically inhibits luciferase activity of constructs containing WT 3'-UTR predicted target sequences. Point mutations (mut) in miR-181a/b binding sites abolish luciferase repression in all cases apart from *PPARGC1A*. Data are normalized to negative mimic transfection (dashed line). $N = 6$ independent experiments.

C  qPCR reveals upregulation of miR-181a/b targets in the eyes of *miR-181a/b-1$^{-/-}$* versus *miR-181a/b-1$^{+/+}$* animals. $n \geq 5$ animals/genotype.

D  *miR-181a/b-1$^{-/-}$* mice show increased mtDNA content versus *miR-181a/b-1$^{+/+}$* mice as measured by qPCR. $N = 4$ animals/genotype.

E  WB analysis (left panel) reveals increased levels of mitochondrial proteins in the eyes of *miR-181a/b-1$^{-/-}$* (-/-) versus *miR-181a/b-1$^{+/+}$* (+/+) mice (quantified in the right panel). Data are normalized to either p115 or Gapdh. $N \geq 3$ animals/genotype. Please note that all compared bands from +/+ and −/− samples are from the same blots, which were cropped and shown split for the sake of data presentation clarity.

F  WB analysis on mitochondrial and cytosolic fractions (left panel) shows increased levels of p62 and Parkin in mitochondrial fraction from the eye of −/− versus +/+ mice (quantified in the right panel). Data are normalized to Ndufb8 or Gapdh for mitochondrial and cytosolic fractions, respectively. $N = 3$ animals/genotype.

G  Cell death analysis shows that miR-181a/b silencing protects SH-SY5Y cells from FCCP treatment. $N \geq 7$ independent experiments.

Data information: *P*-values were calculated by one-tailed Student's *t*-test in (A–C, E and F), by two-tailed Student's *t*-test in (D), and by two-way ANOVA with *post hoc* Tukey's analysis in (G); error bars are SEM.

Source data are available online for this figure.

## miR-181a/b inhibition protects neurons from cell death and ameliorates the phenotype of *in vivo* models of MLS syndrome

Increased mitochondrial biogenesis and clearance were previously shown to exert protective effects in mitochondrial dysfunction (Viscomi *et al*, 2011; Johnson *et al*, 2013; Cerutti *et al*, 2014; Civiletto *et al*, 2015, 2018). We therefore decided to verify whether miR-181a/b inactivation could protect cells from mitochondrial damage. First, we verified whether miR-181a/b downregulation protects SH-SY5Y neuron-like cells from FCCP, a potent OXPHOS uncoupler. In control cells, 6 hours (h) of FCCP treatment induced a significant increase in the extent of cell death. Interestingly, miR-181a/b-silenced cells did not display any differences in cell death after FCCP treatment (Fig 1G), indicating that miR-181a/b silencing protects cells from mitochondrial damage.

Based on the above results, we decided to evaluate the neuroprotective effect of miR-181a/b inactivation in *in vivo* models of MDs. Toward this goal, we examined the consequences of miR-181a/b downregulation in two fish models for a rare inherited form of MD, the MLS syndrome. MLS is a neurodevelopmental disorder characterized by microphthalmia, brain abnormalities, and skin defects in heterozygous females and *in utero* lethality in hemizygous males (Indrieri & Franco, 2016). The disease is due to mutations in key players of the MRC, such as the holocytochrome c-type synthase (*HCCS*), involved in complex III function (Bernard *et al*, 2003; Wimplinger *et al*, 2006; Indrieri *et al*, 2013), and *COX7B*, the 7B subunit of cytochrome c oxidase (MRC complex IV) (Indrieri *et al*, 2012). We previously generated two medakafish (*Oryzias latipes*) models of MLS by knocking down, using a Morpholino(MO)-based approach, *hccs* or *cox7B* expression (Indrieri *et al*, 2012, 2013). Both models (*hccs*-MO and

*cox7B*-MO) showed a severe microphthalmic and microcephalic phenotype due to increased cell death in the CNS (Indrieri *et al*, 2012, 2013, 2016). In medaka, the mature forms of miR-181a and miR-181b are perfectly conserved with respect to their mammalian counterparts, in terms of both sequence identity (100%) and pattern of expression in the retina and brain (Carrella *et al*, 2015). We found that MO-mediated silencing of miR-181a/b in medaka leads to increased levels of the majority of targets involved in mitochondrial biogenesis and function, and in autophagy (Fig EV2). Interestingly, downregulation of miR-181a/b in either of the above-mentioned MLS medaka models led to a notable reduction of cell death in the eye and brain, as shown by TUNEL and caspase activation assays (Fig 2A–C). Accordingly, miR-181a/b downregulation resulted in full rescue of the disease phenotype in about 50% of both *hccs* (Fig 3A–C and M) and *cox7B* morphants (Fig 3G–I and N). Notably, MO-mediated silencing of miR-181a/b did not cause any obvious morphological alteration in the controls (Fig EV3A and B). These data show that the downregulation of miR-181a/b ameliorates the phenotype in both MRC complex III and IV defective models, indicating that the protective effect of miR-181a/b silencing is gene-independent.

Current evidence supports the protective role of autophagy and mitophagy in MDs as well as in other neurodegenerative disorders associated with impairment of mitochondrial functions (Lightowlers *et al*, 2015; Viscomi *et al*, 2015; Civiletto *et al*, 2018). Since we showed that miR-181a/b regulate mitophagy and autophagy in the eye, we tested whether the latter processes are implicated in the amelioration of the phenotype mediated by miR-181a/b downregulation. For this purpose, we treated *hccs*-MO/miR-181a/b-MO- and *cox7B*-MO/miR-181a/b-MO-injected embryos with Bafilomycin A1 (Baf-A1), a general autophagy inhibitor, at 50 nM. This concentration is able to block autophagy without inducing any obvious morphological abnormality in control embryos (Fig EV3C and F). Interestingly, Baf-A1 treatment abolished the protective effect of miR-181a/b downregulation in a significant number of both *hccs*-MO/miR-181a/b-MO- (Fig 3D and M) and *cox7B*-MO/miR-181a/b-MO-injected embryos (Fig 3J and N), indicating that increased autophagy/mitophagy contributes to the phenotype amelioration observed in MLS medaka models.

Since miR-181a/b downregulation also increased the mRNA levels of *erk2*, a key component of the MAPK/ERK cascade, and of *bcl2* and *mcl1*, members of the Bcl2 anti-apoptotic family (Fig EV2), we also tested whether the latter two pathways were involved in the amelioration of the MLS phenotype. Therefore, we treated *hccs*-MO/miR-181a/b-MO- and *cox7B*-MO/miR-181a/b-MO-injected embryos with PD98059, a selective inhibitor of the MAPK/ERK pathway (Alessi *et al*, 1995), or with HA14-1, an inhibitor of Bcl-2 proteins (Wang *et al*, 2000). PD98059 and HA14-1 were used at concentrations that did not induce any morphological alterations in control embryos [(Carrella *et al*, 2015) and Fig EV3D and E]. PD98059 treatment did not have any effect on the extent of phenotypic rescue in neither MLS models (Fig 3E, K, M and N), suggesting that the MAPK pathway is not primarily involved in the protective effect exerted by miR-181a/b downregulation in the analyzed models. On the other hand, HA14-1 significantly reduced the extent of phenotypic rescue in *hccs*-MO/miR-181a/b-MO-injected embryos, but not in *cox7B*-MO/miR-181a/b-MO-injected embryos (Fig 3F and L–N).

Taken together, these data suggest that increased autophagy/mitophagy are involved in the amelioration of the MLS phenotype in both models, indicating a primary role for this pathway in miR-181a/b downregulation-mediated neuronal protection. Moreover, increased levels of *bcl2* play a role in the amelioration of *hccs*-defective embryos only, indicating that the contribution of the Bcl2-mediated apoptotic pathway may vary depending on the specific mitochondrial defect.

## miR-181a/b inactivation ameliorates the phenotype of a drug-induced mouse model of LHON

To validate the efficacy of miR-181a/b inactivation in mammalian models of MDs as well, we first exploited a drug-induced mouse model of LHON, a non-syndromic form of mitochondrial optic neuropathy characterized by degeneration of retinal ganglion cells (RGCs) that leads to loss of central vision. LHON represents one of the most frequent forms of MDs with a prevalence of 1:30,000 individual with an onset occurring between the ages of 15 and 35 (Carelli *et al*, 2004). Similar to other MDs, LHON is characterized by high genetic heterogeneity being caused by mutations in multiple genes (Carelli *et al*, 2004; Meyerson *et al*, 2015). In about 95% of LHON patients, mutations are located in the mitochondrially encoded *ND1*, *ND4*, or *ND6* genes, which encode complex I subunits (Carelli *et al*, 2004; Meyerson *et al*, 2015). Intravitreal injection of rotenone, an MRC complex I inhibitor, leads to the damage of RGCs, and injected mice are considered a reliable drug-induced LHON model (Carelli *et al*, 2013). In order to assess whether the genetic inactivation of miR-181a/b exerts a protective effect against rotenone-induced LHON, *miR-181a/b-1*$^{+/+}$ and *miR-181a/b-1*$^{-/-}$ mice were intravitreally injected with rotenone unilaterally. As internal controls, contralateral eyes were injected with DMSO. As previously reported (Heitz *et al*, 2012), *miR-181a/b-1*$^{+/+}$ control mice injected with rotenone displayed a notable reduction in the number of RGCs (Fig 4A and B). In contrast, no difference was observed in the number of RGCs between rotenone- and DMSO-injected retina in *miR-181a/b-1*$^{-/-}$ mice, both 1 and 2 weeks after injection (Fig 4A and B). Interestingly, rotenone-injected *miR-181a/b-1*$^{-/-}$ mice also showed preserved MRC complex I activity in RGCs, as demonstrated by the staining of NADH dehydrogenase activity, with respect to rotenone-injected *miR-181a/b-1*$^{+/+}$ mice (Fig 4C). Accordingly, CoxIV immunofluorescence analysis showed that *miR-181a/b-1* depletion protects mitochondrial network integrity from rotenone-induced damage (Fig 4D). Finally, we also tested retinal function by optokinetic response (OKR). For this analysis, *miR-181a/b-1*$^{+/+}$ and *miR-181a/b-1*$^{-/-}$ mice were bilaterally injected with rotenone or DMSO. Interestingly, *miR-181a/b-1*$^{+/+}$ mice showed a severe decrease in visual performance starting 1 week after rotenone administration, whereas *miR-181a/b-1*$^{-/-}$ mice were significantly protected against rotenone-induced toxicity (Fig 4E). Ablation of *miR-181a/b-1* rescued visual dysfunction even 2 weeks after rotenone injection (Fig 4E). Importantly, *miR-181a/b-1* ablation did not cause *per se* neither abnormalities in retinal morphology and RGC number [as shown by immunofluorescence analysis with different retinal cell markers (Fig EV4A–J and M)] nor alteration of retinal function [as assessed by electroretinogram (ERG) analysis (Fig EV4K and L)]. Altogether, these data indicate that miR-181a/

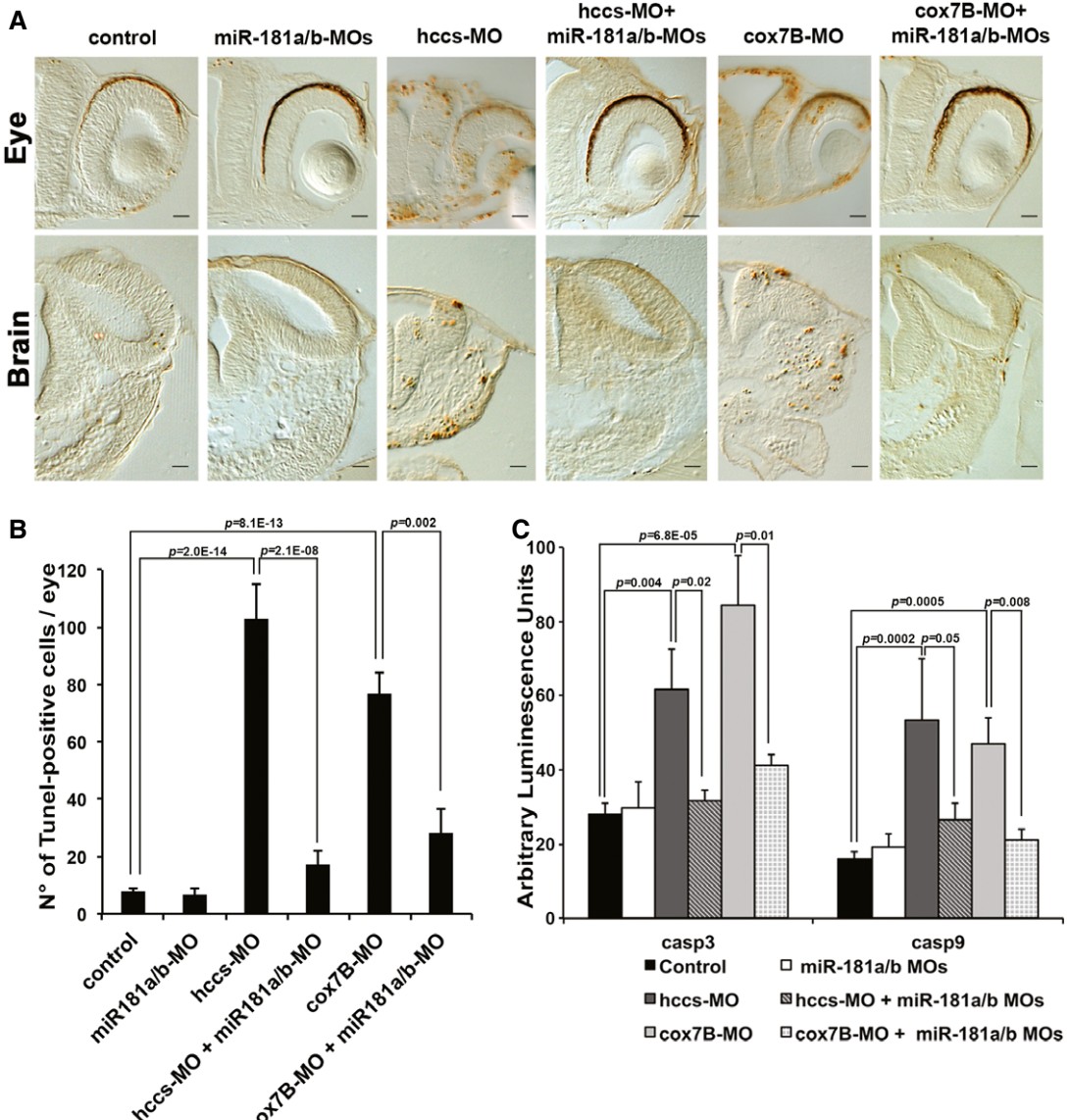

**Figure 2. miR-181a/b inhibition counteracts cell death in MLS medakafish models.**

A  TUNEL assays on eye and brain sections of stage (st)30 medakafish control and MO-injected embryos reveal a decrease in the number of apoptotic cells in both *hccs*-MO/miR-181a/b-MO- and *cox7B*-MO/miR-181a/b-MO-injected compared to *hccs*-MO- and *cox7B*-MO-injected embryos. Scale bars are 20 μm.

B  TUNEL-positive cells/eye. *n* ≥ 5 eyes for each model.

C  Caspase assays show restored levels of caspase-3 and caspase-9 activities in *hccs*-MO/miR-181a/b-MO- and *cox7B*-MO/miR-181a/b-MO-injected embryos with respect to *hccs*-MO- and *cox7B*-MO-injected embryos. *n* ≥ 5 embryos for each model.

Data information: *P*-values were calculated by analysis of deviance for negative binomial generalized linear model in (B) and by one-way ANOVA with *post hoc* Tukey's analysis in (C); error bars are SEM.

b silencing exerts a protective effect in a drug-induced mammalian model of LHON and therefore represents a potential therapeutic strategy for this condition.

### miR-181a/b inactivation rescues the LHON phenotype in the *Ndufs4*⁻/⁻ mouse model

Rotenone-injected mice are considered a reliable model of LHON (Carelli *et al*, 2013). However, the latter represents an acute model in

which it is difficult to dissect the mechanism by which miR-181a/b downregulation exerts its protective function. Therefore, we decided to exploit the $Ndufs4^{-/-}$ mouse model (Kruse *et al*, 2008) to further characterize and validate the efficacy of miR-181a/b inactivation in mammalian models of MDs. Homozygous mutations in *NDUFS4* are responsible for one of the most severe forms of Leigh syndrome (LS), an inherited neurodegenerative condition, characterized by poor prognosis as patients typically die before 3 years of age. NDUFS4 is a nuclear DNA-encoded MRC complex I subunit located in the NADH

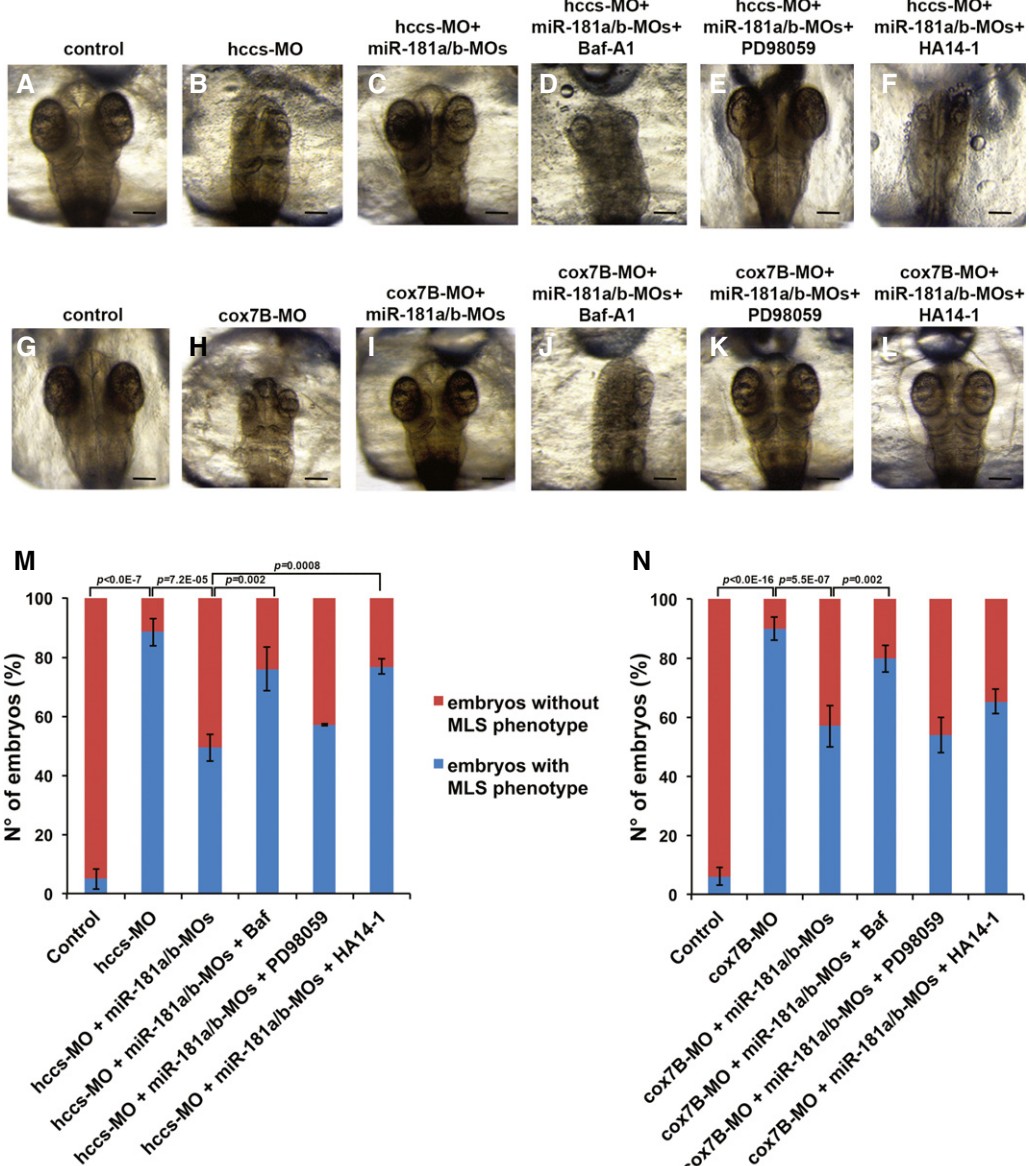

**Figure 3.  miR-181a/b downregulation ameliorates the phenotype of MLS medakafish models.**

A–L    Representative images of st30 medaka embryos injected with *hccs*-MO (B) and *cox7B*-MO (H) alone or co-injected with miR-181a/b-MOs (C, I). Co-injection of miR-181a/b-MOs rescues microphthalmia and microcephaly in both *hccs*-MO and *cox7B*-MO embryos. (D–F, J–L) *hccs*-MO/miR-181a/b-MO- and *cox7B*-MO/miR-181a/b-MO-injected embryos were treated with Baf-A1, PD98059, or HA14-1. Baf-A1 treatment counteracts the protective effect of miR-181a/b downregulation in both *hccs*-MO/miR-181a/b-MO and *cox7B*-MO/miR-181a/b-MO embryos (D, J). PD98059 treatment does not interfere with the modulation of the MLS phenotype mediated by miR-181a/b downregulation (E, K). HA14-1 treatment counteracts the effect of miR-181a/b downregulation in the *hccs*-MO/miR-181a/b-MO model but not in the *cox7B*-MO/miR-181a/b-MO embryos (F, L). Scale bars are 100 μm.

M, N    Percentage of embryos with or without MLS phenotype in the different conditions illustrated in (A–L) in *hccs*-MO (M) and *cox7B*-MO (N) embryos. $N \geq 300$ embryos/conditions. *P*-values were calculated by analysis of deviance for generalized linear model; error bars are SEM.

dehydrogenase domain of the complex, responsible for its catalytic activity (Breuer *et al*, 2013; Stroud *et al*, 2016; Zhu *et al*, 2016). Mice with homozygous *Ndufs4* inactivation develop a severe encephalopathy resembling LS that results in death between 50 and 60 postnatal (P) days (Kruse *et al*, 2008; Breuer *et al*, 2013). In addition, these mice also show a LHON phenotype characterized by RGC dysfunction starting from P16 to P20 and leading to a detectable RGC death after P42 (Yu *et al*, 2015; Song *et al*, 2017), which makes the *Ndufs4*

knockout line a good model to study the effect of miR-181a/b inactivation in LHON. By crossing *miR-181a/b-1*$^{-/-}$ with *Ndufs4*$^{-/-}$ mice, we obtained *Ndufs4*$^{-/-}$/*miR-181a/b-1*$^{-/-}$ animals that we exploited to study the effect of miR-181a/b inactivation on the retinal phenotype. As previously reported (Yu *et al*, 2015), *Ndufs4*$^{-/-}$ mice showed a significant reduction in the number of RGCs at P55, as confirmed by NeuN-positive cell count (Fig 5A, B and E). This reduction was rescued in *Ndufs4*$^{-/-}$/*miR-181a/b-1*$^{-/-}$ retinas (Fig 5C and E),

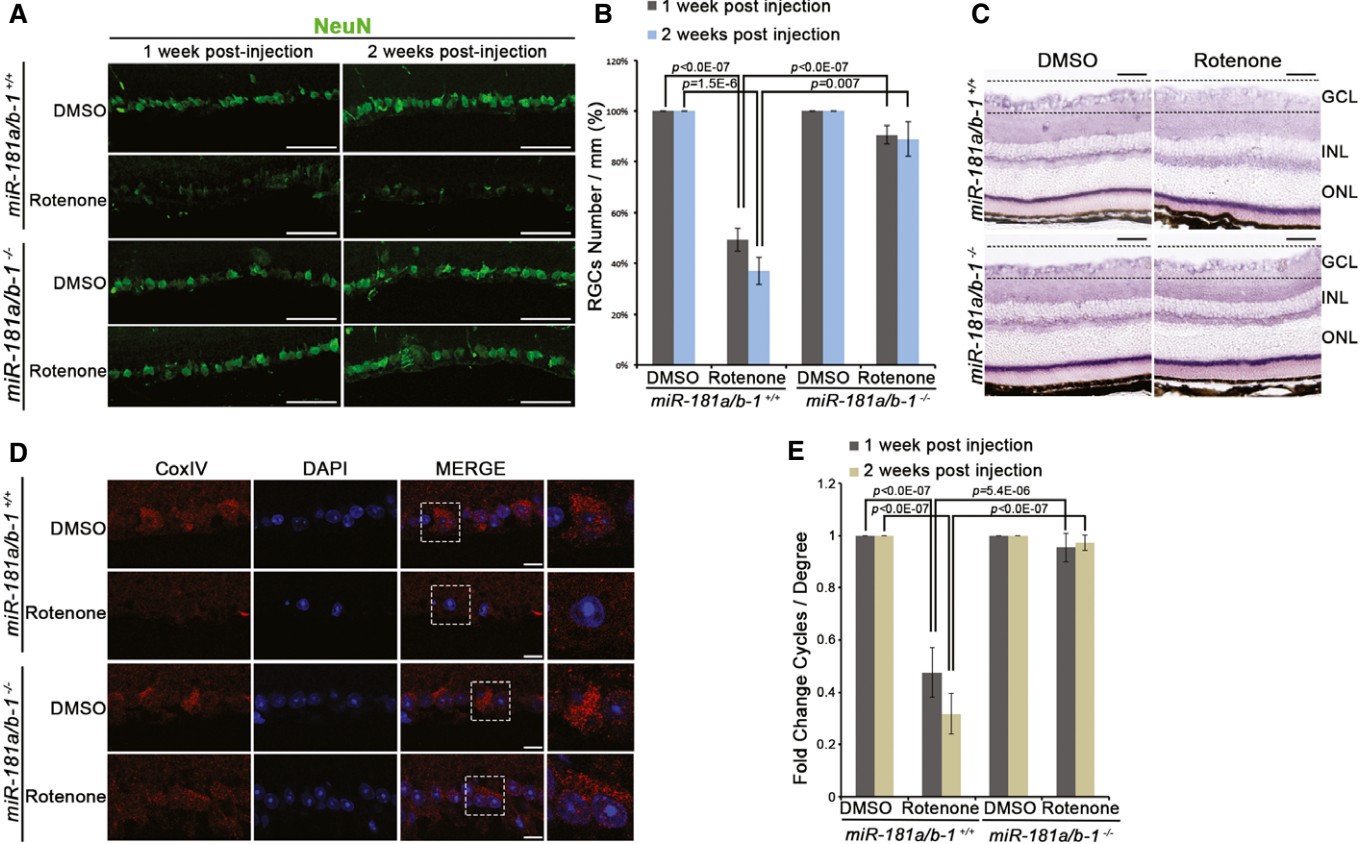

**Figure 4. Inactivation of miR-181a/b protects RGCs from cell death in a rotenone-induced mouse model of LHON syndrome.**

A   Immunofluorescence analysis with anti-NeuN antibody in the retina of miR-181a/b-1[+/+] and miR-181a/b-1[−/−] mice intravitreally injected with rotenone or DMSO. RGCs are preserved in miR-181a/b-1[−/−] rotenone-injected mice with respect to controls at both one and two weeks after injection. Scale bars are 50 μm.

B   Number of RGCs/mm (indicated as %) at 1 and 2 weeks post-injection. N = 10.

C   NADH dehydrogenase histochemical reaction on retinal sections of miR-181a/b-1[+/+] and miR-181a/b-1[−/−] mice intravitreally injected with rotenone or DMSO. At 1 week post-injection, NADH dehydrogenase activity is lost in RGCs (GCL, areas within dashed lines) of miR-181a/b-1[+/+] rotenone-injected eyes, while it is preserved in those of miR-181a/b-1[−/−] rotenone-injected eyes. GCL, ganglion cell layer; INL, inner nuclear layer; ONL, outer nuclear layer. Scale bars are 50 μm.

D   Immunofluorescence analysis, with an anti-CoxIV antibody, in the retina of miR-181a/b-1[+/+] and miR-181a/b-1[−/−] mice, injected with either rotenone or DMSO, showed preserved mitochondria in miR-181a/b-1[−/−] rotenone-injected eyes at 1 week post-injection. Dashed boxes indicate the area of magnifications shown in right panels. Scale bars are 10 μm.

E   Graphical representation of the results of the optokinetic tracking assays reported as fold change of cycles/degree. Visual acuity is preserved in miR-181a/b-1[−/−] rotenone-injected mice with respect to controls at both 1 and 2 weeks after injection. N = 10.

Data information: P-values were calculated by one-way ANOVA with *post hoc* Tukey's analysis; error bars are SEM.

in which the number of NeuN-positive cells was indistinguishable from both WT and miR-181a/b-1[−/−] mice (Fig 5D and E). Moreover, the OKR test showed a significant amelioration of visual acuity in Ndufs4[−/−]/miR-181a/b-1[−/−] compared to Ndufs4[−/−] mice (Fig 5F) at P25, indicating a remarkable effect of miR-181a/b downregulation on both RGC dysfunction and death. Finally, we also analyzed the ERG response, previously reported to be altered in Ndufs4[−/−] mice with the absence of b-wave response (Kruse *et al*, 2008; Yu *et al*, 2015). We observed a significant amelioration of the b-wave (both in scotopic and in photopic ranges) in Ndufs4[−/−]/miR-181a/b-1[−/−] with respect to Ndufs4[−/−] mice (Fig 5G and H), indicating a significant improvement of retinal response to light stimuli at P30. Altogether, these results indicate that in the Ndufs4[−/−] model miR-181a/b downregulation preserves RGCs from death and ameliorates the visual function before the miRNAs exert their activity on RGC death.

To gain insight into the molecular mechanisms underlying the amelioration of the miR-181a/b inhibition-induced Ndufs4[−/−] retinal phenotype, we analyzed the expression levels of miR-181a/b targets in Ndufs4[−/−]/miR-181a/b-1[−/−] mice and we observed a significant increase in the Nrf1 and Park2 transcripts as compared to Ndufs4[−/−] (Fig EV5A). As previously mentioned, Park2 and Nrf1 are important players in the control of, respectively, mitophagy (Narendra *et al*, 2008) and mitochondrial biogenesis (Wu *et al*, 1999; Finck & Kelly, 2006), i.e., pathways that we showed to be upregulated in miR-181a/b-1[−/−] versus WT mice (Fig 1). We therefore hypothesized that the activation of these pathways significantly contributes to the phenotypic improvement observed in Ndufs4[−/−]/miR-181a/b-1[−/−] mice. Notably, the levels of Parkin and p62 were increased in mitochondrial fractions from Ndufs4[−/−]/miR-181a/b-1[−/−] with respect to Ndufs4[−/−] mice (Fig EV5C and D) indicating enhanced mitophagy

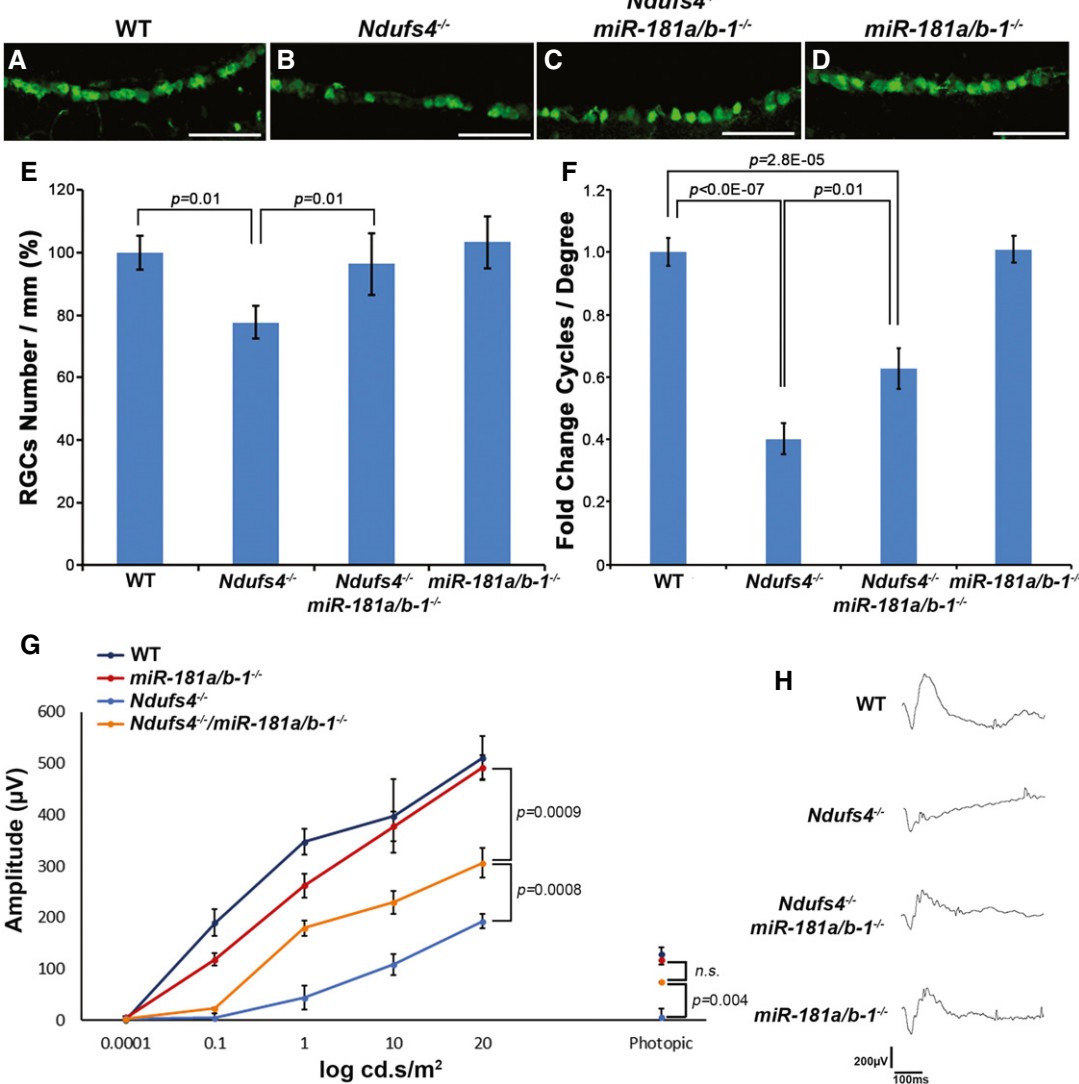

**Figure 5. miR-181a/b depletion protects RGCs and ameliorates visual function in *Ndufs4*$^{-/-}$ mice.**

A–D    Immunofluorescence analysis with anti-NeuN antibody in the retina of *WT*, *Ndufs4*$^{-/-}$, *Ndufs4*$^{-/-}$/*miR-181a/b-1*$^{-/-}$, and *miR-181a/b-1*$^{-/-}$ mice. RGCs are preserved in *Ndufs4*$^{-/-}$/*miR-181a/b-1*$^{-/-}$ with respect to *Ndufs4*$^{-/-}$ mice. Scale bars are 50μm.

E    Number of RGCs/mm (%). $N = 5$.

F    Graphical representation of the results of the optokinetic tracking assays reported as fold change of cycles/degree. Visual acuity is preserved in *Ndufs4*$^{-/-}$/*miR-181a/b-1*$^{-/-}$ with respect to *Ndufs4*$^{-/-}$ mice. $N \geq 8$.

G, H    Electroretinographic analysis reveals amelioration of b-wave patterns in *Ndufs4*$^{-/-}$/*miR-181a/b-1*$^{-/-}$ with respect to *Ndufs4*$^{-/-}$ mice. $N \geq 5$.

Data information: *P*-values were calculated by one-way ANOVA with *post hoc* Tukey's analysis in (E and F) and by two-way ANOVA repeated measures with *post hoc* analysis in (G). *P*-values in (G) refer to 20 log cd.s/m$^2$ and photopic points. *P*-values for the other points are reported in Appendix Table S1; error bars are SEM.

activation in response to miR-181a/b depletion. Electron microscopy (EM) studies showed lack of accumulation of autophagosomes/autolysosomes containing undigested material excluding a possible mitophagy block at the level of autophagosome recruitment or autophagosome/lysosome fusion (Fig 6B). Moreover, consistent with *Nrf1* upregulation, we showed by EM increased number of mitochondria at both P30 and P55 in RGCs of *Ndufs4*$^{-/-}$/*miR-181a/b-1*$^{-/-}$ with respect to *Ndufs4*$^{-/-}$ mice (Fig 6A–C). The augmented mitochondrial biogenesis was also confirmed by increase of mtDNA in *Ndufs4*$^{-/-}$/*miR-181a/b-1*$^{-/-}$ versus *Ndufs4*$^{-/-}$ mice, as measured

by qPCR (Fig 6D). Interestingly, the increase in number of mitochondria in RGCs of *Ndufs4*$^{-/-}$/*miR-181a/b-1*$^{-/-}$ mice was accompanied by a notable amelioration of their morphological structure, such as increased number of cristae and more electron-dense matrix, with respect to *Ndufs4*$^{-/-}$ mice (Fig 6A and B). Notably, *NRF1* downregulation significantly abolished the miR-181a/b-mediated rescue in SH-SY5Y cells treated with FCCP, indicating that indeed NRF1 is implicated in the neuroprotective effect exerted by miR-181a/b downregulation (Fig EV5B). Finally, we also analyzed the biochemical activity of MRC complexes I, II, and IV. The latter analyses showed an

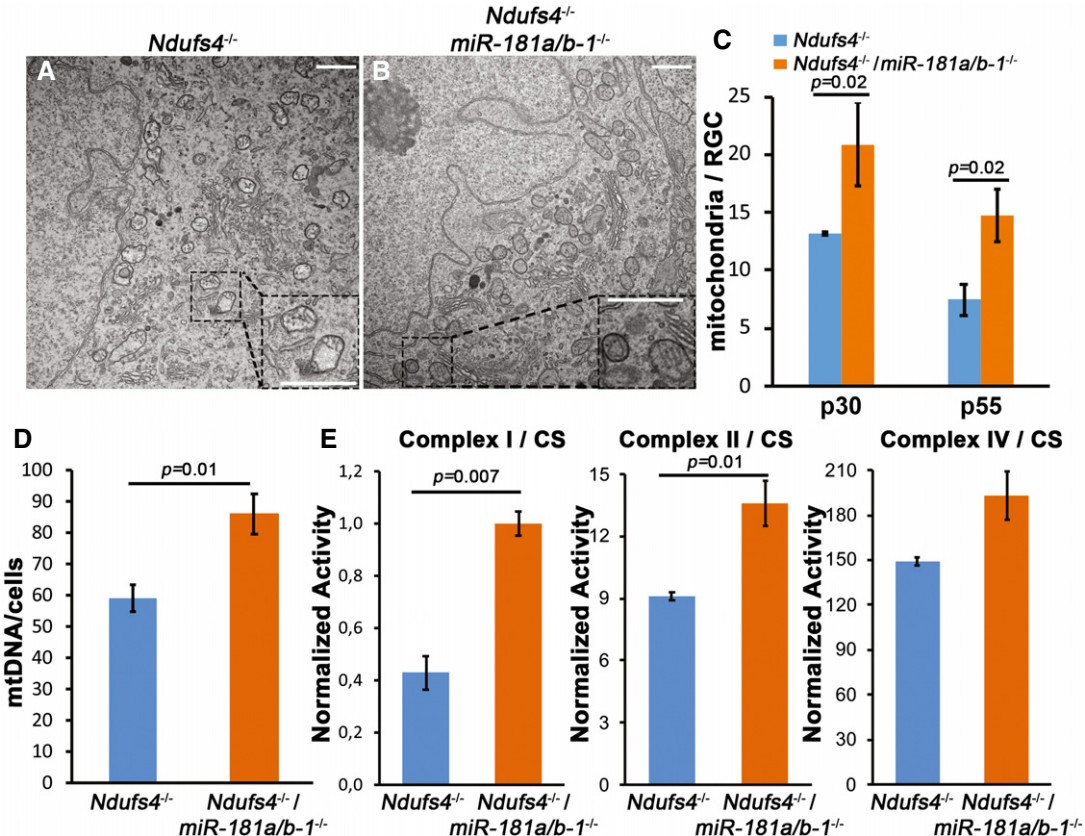

**Figure 6. miR-181a/b depletion enhances mitochondrial biogenesis Ndufs4$^{-/-}$ mice.**

A–C Electron microscopy analysis shows amelioration of mitochondrial morphology and increase of mitochondrial number in Ndufs4$^{-/-}$/miR-181a/b-1$^{-/-}$ versus Ndufs4$^{-/-}$ mice. Scale bars are 1 μm. The quantitative increase in mitochondria is reported in (C) as number of mitochondria/RGC, at p30 and p55. N ≥ 2 animals/genotype.

D Ndufs4$^{-/-}$/miR-181a/b-1$^{-/-}$ mice show increased mtDNA content versus Ndufs4$^{-/-}$ mice as measured by qPCR. N ≥ 4 animals/genotype.

E Biochemical activity of MRC complexes I, II, and IV normalized by the percentage of citrate synthase (CS) activity in Ndufs4$^{-/-}$/miR-181a/b-1$^{-/-}$ versus Ndufs4$^{-/-}$ mice. N ≥ 3 animals/genotype.

Data information: P-values were calculated by two-tailed Student's t-test; error bars are SEM.

increase in complex I and II activities, and a similar trend for complex IV activity, normalized to the activity of the Krebs cycle enzyme citrate synthase in the Ndufs4$^{-/-}$/miR-181a/b-1$^{-/-}$ mice compared with the Ndufs4$^{-/-}$ mice at P30 (Fig 6E), indicating that miR-181a/b downregulation is partially able to counteract the complex I biochemical defect of the MRC in the Ndufs4$^{-/-}$ mouse.

Altogether, these data indicate that miR-181a/b downregulation exerts its protective effect mainly through the simultaneous stimulation of mitophagy and mitochondrial biogenesis, which leads to enhancement of mitochondrial turnover in the retina that ultimately results in amelioration of the mitochondrial morphological and biochemical phenotypes (Fig 7).

## Discussion

In this report, we provide a proof of principle for the genetic inactivation or downregulation of miR-181a/b exerting a protective effect from mitochondrial-mediated neurodegeneration in both *in vitro*

and *in vivo* MD models, regardless of the underlying etiopathogenetic events thus highlighting these two miRNAs as potential, gene-independent, therapeutic targets for MDs characterized by neuronal degeneration.

We found that miR-181a/b control genes implicated in mitochondrial biogenesis, functionality, and antioxidant response. In particular, our data demonstrate that these miRNAs control mitochondrial biogenesis in the retina by direct targeting of *Nrf1* and, together with the already known effect on general autophagy and mitophagy (Ouyang *et al*, 2012; Tekirdag *et al*, 2013; Cheng *et al*, 2016), uncover a fundamental role of miR-181a/b in the control of mitochondrial turnover in the CNS. Overall, these data point to miR-181a/b as possible hubs in the gene pathways underlying mitochondrial homeostasis in both physiological and pathological conditions. We propose that the broad and beneficial impact of miR-181a/b downregulation on MD models relies on the ability of these two miRNAs to simultaneously and finely modulate the above-mentioned mitochondrial-related pathways, as opposed to more drastic and individual modulations of each of them.

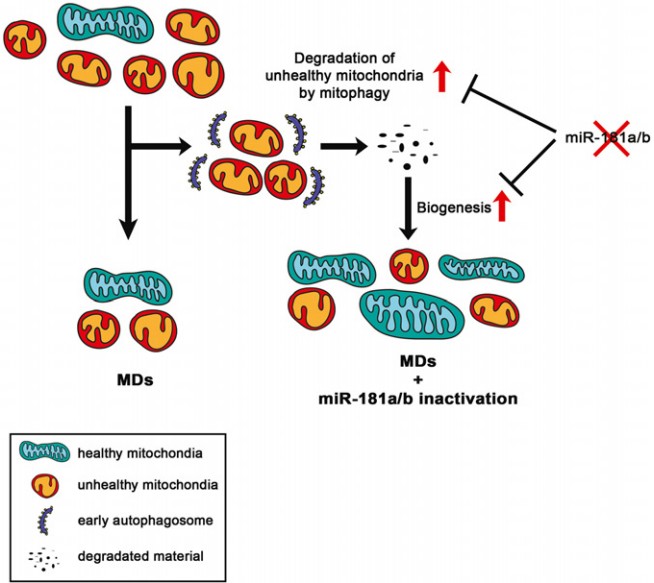

**Figure 7. Proposed model for the mechanism of action of miR-181a/b inactivation in MDs.**

In MDs, a portion of unhealthy mitochondria is degraded by mitophagy resulting in a decreased number of mitochondria, which is not efficiently balanced by mitochondrial biogenesis. Inactivation of miR-181a/b leads to the simultaneous increase in mitophagy and mitochondrial biogenesis (red arrows), inducing a significant enhancement of mitochondrial turnover and activity and ensuring a more efficient protection from cell death.

Increased autophagy and mitophagy have been shown to exert protective effects in MDs and in other neurodegenerative diseases (Decressac et al, 2013; Martinez-Vicente, 2017). Of note, Parkin is one of the main players in the clearance of damaged mitochondria via the autophagy pathway (Geisler et al, 2010; Vives-Bauza et al, 2010). Moreover, treatment with the mTOR inhibitor rapamycin, which activates autophagy, significantly delayed both neurodegeneration progression and the fatal outcome of the $Ndufs4^{-/-}$ mouse (Johnson et al, 2013) and of a mouse model of complex IV deficiency (Civiletto et al, 2018), although a general consensus on the mechanisms underlying the effects of rapamycin on primary mitochondrial dysfunction is still lacking. Here, we show that downregulation of miR-181a/b leads to increased autophagy and mitophagy in the retina. In line with these data, inhibition of general autophagy in both hccs/miR-181a/b-MOs- and cox7B/miR-181a/b-MOs-injected embryos reduces the neuroprotective effect of miR-181a/b downregulation on the MLS phenotype, indicating that increased autophagy/mitophagy significantly contributes to the phenotypic rescue in both models. Moreover, analysis of $Ndufs4^{-/-}/miR-181a/b-1^{-/-}$ mice shows increased mitophagy in the retina, confirming the importance of the activation of this process in the amelioration of MD neurodegenerative phenotypes in response to miR-181a/b downregulation. In addition, Parkin overexpression may be beneficial for neuronal cell survival by acting on other pathways such as mitochondrial dynamics, through enhancement of proteasomal activity (Khandelwal & Moussa, 2010; Rana et al, 2013) and transcriptional control of key life/death genes such as p53 (Alves da Costa et al, 2018).

Although the removal of dysfunctional mitochondria would limit the risk of apoptosis, an uncontrolled activation of mitophagy without an appropriate compensatory mitochondrial biogenesis may contribute to further mitochondrial dysfunction resulting in loss of mitochondrial mass, energetic collapse, and cell death (Zhu et al, 2013; Villanueva Paz et al, 2016). Furthermore, enhanced mitochondrial biogenesis could be per se protective in MDs. Variability in mitochondrial DNA content and mitochondrial biogenesis is indeed associated with incomplete penetrance in non-manifesting carriers of LHON (Giordano et al, 2014), and increased biogenesis was found to improve the phenotype of different in vitro and in vivo MD models (Komen & Thorburn, 2014). However, some discrepant and contradictory results obtained using compounds that trigger mitochondrial biogenesis (including bezafibrate, resveratrol, and AICAR) (Komen & Thorburn, 2014) clearly indicate that additional work is needed to effectively exploit modulation of mitochondrial biogenesis to increase mitochondrial functionality in patients. The tight balance between mitophagy and mitochondrial biogenesis is emerging as a critical aspect in the determination of cell viability in diseases characterized by mitochondrial dysfunction (Zhu et al, 2013). Here, we show that miR-181a/b can simultaneously control both processes. Concomitantly to the increased mitophagy observed following miR-181a/b inactivation, we detected increased levels of several mitochondrial proteins and of mtDNA, indicating enhanced mitochondrial biogenesis. Accordingly, miR-181a/b-depleted animals showed enhanced MRC complex I activity and preserved mitochondrial integrity in the rotenone-induced LHON model. Interestingly, consistent with the higher levels of Nrf1, the $Ndufs4^{-/-}/miR-181a/b-1^{-/-}$ model showed a significantly increased number of mitochondria in the retina compared to $Ndufs4^{-/-}$ mice indicating enhanced mitochondrial biogenesis. Moreover, $Ndufs4^{-/-}/miR-181a/b-1^{-/-}$ mice show amelioration of mitochondrial morphological structure with respect to $Ndufs4^{-/-}$ mice. In addition, analysis of the biochemical activity of MRC complexes showed an increase in the activity of complex I in $Ndufs4^{-/-}/miR-181a/b-1^{-/-}$ compared with $Ndufs4^{-/-}$ mice. This increase, although significant, is still very far from compensating the profound complex I deficiency shown by the $Ndufs4^{-/-}$ mice. However, in $Ndufs4^{-/-}/miR-181a/b-1^{-/-}$ animals, we have observed a concomitant increase in the activity of complex II, which contributes to the amelioration of the phenotype by increasing the electron flux through complexes III and IV. We hypothesize that this enhancement of complex activities is mediated by miR-181a/b action on degradation of the most dysfunctional mitochondria and enhanced mitochondrial biogenesis (proposed in Fig 7), which translates in higher MRC activities.

We cannot exclude that other mitochondrial-related gene pathways may contribute to the miR-181a/b downregulation effect. For example, another important pathway modulated by miR-181a/b is the mitochondrial-dependent cell death in which the Bcl-2 protein family has a key regulatory role (Rasola & Bernardi, 2007; Tait & Green, 2010). Although the contribution of apoptosis to the pathogenesis of primary MDs is not univocally established, increasing the levels of anti-apoptotic Bcl-2 proteins may represent an effective strategy to prolong cell survival and slow down disease progression. Notably, overexpression of the Bcl-2 family protein Bcl-xL counteracted increased cell death and ameliorated the phenotype of the hccs-deficient MLS medaka model (Indrieri et al, 2013). Here, we show that miR-181a/b inhibition leads to increased levels of Bcl-2

family members, such as Bcl-2 and Mcl-1, and rescues the phenotype of both MLS models tested. Moreover, HA14-1, a potent inhibitor of Bcl-2 proteins (Wang *et al*, 2000), significantly abolishes miR-181a/b-mediated amelioration of the MLS phenotype in the hccs-defective, but not in the cox7B-defective model. These data indicate that a single pathway, e.g., mitochondrial-dependent cell death, and its modulation, may have different effects on highly related forms of MDs, depending on the underlying genetic defect. In addition, our data indicate that the amelioration of the phenotype observed in *Ndufs4*$^{-/-}$ mice upon miR-181a/b downregulation preceded the effect on RGC death which is detectable only after P42. We showed that miR-181a/b depletion improves mitochondrial number, morphology, and MRC complex activities and rescues the visual function before P30, thus indicating that the beneficial effects are mainly due to mitochondrial turnover enhancement rather than activation of the Bcl2-mediated cell death pathway. This observation strengthens our hypothesis that the simultaneous modulation of more than one pathway represents a more effective strategy to treat MDs.

Nevertheless, additional efforts are necessary for the exhaustive dissection of the molecular mechanisms underlying the remarkable efficacy of miR-181a/b downregulation in MDs, and to evaluate its safety. It is to be noted that *miR-181a/b-1*$^{-/-}$ mice show normal life-span (Henao-Mejia *et al*, 2013) and no obvious abnormalities in the eyes (Fig EV4), which provides initial support to the safety of the downregulation of these two miRNAs.

MDs are genetically heterogeneous disorders. Although each individual disorder is rare, collectively the total prevalence of adult MDs is estimated to be of approximately 1 in 4,300 representing one of the most prevalent groups of inherited neurological diseases (Gorman *et al*, 2015). In this study, we demonstrate that downregulation of miR-181a/b exerts a neuroprotective effect in multiple models of MDs. Our results are of particular interest for genetically heterogeneous conditions, such as LHON and LS for which a gene-independent therapeutic strategy is highly desirable. In summary, we believe that the modulation of miR-181a/b may represent effective gene-independent therapeutic targets for MDs characterized by neuronal degeneration.

## Materials and Methods

### Cell lines

HeLa and SH-SY5Y cells were obtained from ATCC and maintained in culture with Dulbecco's modified Eagle's medium (Gibco) or a 1:1 mixture of Dulbecco's modified Eagle's medium and Nutrient Mixture F-12, supplemented with 10% FBS (EuroClone) and 1% penicillin/streptomycin, respectively, as suggested by the vendor.

### Animal studies

Ethical statement: All studies on fish and mice were conducted in accordance with ARRIVE guidelines for animal research and approved by the Italian Ministry of Health, Department of Public Health, Animal Health, Nutrition and Food Safety, in accordance with the law on animal experimentation (article 7; D.L. 116/92; protocol number: 389/2015-PR and 575/2017-PR). All animal

treatments were reviewed and approved in advance by the Institutional Ethics Committee at the Telethon Institute of Genetics and Medicine (Pozzuoli, Italy).

Medakafish (*O. latipes*) of the cab strain were maintained in the in-house TIGEM facility (28°C on a 14/10 h light/dark cycle). The C57BL/6-miR-181a/b$^{-/-}$ mice were provided by Prof. Richard A. Flavell (Department of Immunobiology, Yale University School of Medicine, New Haven, Connecticut, USA). The B6.129S4-*Ndufs4*$^{+/-}$ strain described in Kruse *et al* (2008) was obtained from The Jackson Laboratory and backcrossed with WT C57BL/6 mice. Mice (male and females, 2–8 weeks old) were housed at the TIGEM animal facility (22°C on a 12/12-h light/dark cycle, humidity-controlled) and kept on an *ad libitum* normal diet (VRF1, Special Diet Services) and free access to tap water.

*Ndufs4*$^{+/-}$ mice were crossed with *miR-181a/b-1*$^{-/-}$ animals to generate *Ndufs4*$^{+/-}$/*miR-181a/b-1*$^{+/-}$ mice. To obtain background homogeneity, *Ndufs4*$^{+/-}$/*miR-181a/b-1*$^{+/-}$ mice were then backcrossed with *miR-181a/b-1*$^{+/+}$ or *miR-181a/b-1*$^{-/-}$ derived from the same littermates. *Ndufs4*$^{+/-}$/*miR-181a/b-1*$^{+/+}$ were crossed with *Ndufs4*$^{+/-}$/*miR-181a/b-1*$^{+/+}$ mice to obtain *Ndufs4*$^{-/-}$/*miR-181a/b-1*$^{+/+}$, and *Ndufs4*$^{+/-}$/*miR-181a/b-1*$^{-/-}$ were crossed with *Ndufs4*$^{+/-}$/*miR-181a/b-1*$^{-/-}$ to obtain *Ndufs4*$^{-/-}$/*miR-181a/b-1*$^{-/-}$. WT (*Ndufs4*$^{+/+}$/*miR-181a/b-1*$^{+/+}$) controls were age-matched and obtained both from littermates and separate litters that were co-housed.

### Luciferase assays

Human 3′-UTR sequences containing miR-181a/b predicted binding sites were amplified by PCR and inserted in the pGL3-tk-luciferase vector. Constructs containing mutagenized miR-181a/b binding sites were obtained using the QuikChange Site-Directed Mutagenesis Kit (Stratagene).

Primer sequences are reported in Appendix Table S2. Plasmids were transfected in HeLa cells (PolyFect reagent, Qiagen). After 7 h, cells were transfected with 100 nM of miRIDIAN negative mimic or mimic-miR-181 (Dharmacon) using Interferin (Polyplus). After 24 h, luciferase activities were quantified using Dual-luciferase Reporter Assay (Promega).

### RNA extractions

Tissue samples were processed in QIAzol lysis reagent (Qiagen). Total RNA from tissue and cell samples was extracted using the RNeasy Extraction Kit (Qiagen), according to manufacturer's instructions.

### Quantitative real-time PCR

For qPCR experiments, cDNAs were generated using QuantiTect Reverse Transcription Kit (Qiagen), according to manufacturer's instructions.

Primers for qPCRs were designed to span two different exons to avoid genomic DNA amplification using *in silico* tools (www.basic.northwestern.edu/biotools/oligocalc.html) to predict their melting temperature ($T_m$) and to avoid the possibility of self-annealing or primer dimerization. The specificity of the designed primers was

tested *in silico* using the BLAT or BLAST tool in Genome Browser (https://genome.ucsc.edu/) or Ensembl (http://www.ensembl.org/index.html). Primers were tested as described (Bustin *et al*, 2009). Primer sequences are reported in Appendix Table S3. Quantification data, obtained in qPCRs on cDNAs obtained from different treatments, are expressed in terms of cycle thresholds (Ct). The *HPRT* and *GAPDH* genes were used as endogenous reference controls for experiments. The *Ct* values were averaged for each in-plate technical triplicate. The averaged *Ct* was normalized as difference in Ct values ($\Delta$Ct) between the analyzed mRNAs and each reference gene in each sample analyzed. The $\Delta$Ct values of each sample were then normalized with respect to the $\Delta$Ct values of the control ($\Delta\Delta$Ct). The variation was reported as fold change ($2^{-\Delta\Delta Ct}$). Each plate was performed in duplicate, and all the results are shown as means $\pm$ SEM of at least three independent biological assays.

## SH-SY5Y transfections and FCCP treatment

SH-SY5Y cells were transfected with 100 nM of negative control miRNA inhibitor and miR-181a and miR-181b inhibitors (Dharmacon) using a lipid-based transfection reagent (DharmaFECT, Dharmacon). Upon 72 h of transfection, SH-SY5Y cells were treated with 25 $\mu$M FCCP. After 6 h of FCCP treatment, dead cells were stained with trypan blue and counted with the automated cell counter (LUNA, Logos Biosystems). For double transfection, SH-SY5Y cells were transfected with 100 nM of negative control miRNA inhibitor or miR-181a and miR-181b inhibitors, and with 100 nM control siRNA or siNRF1 (Dharmacon) using Lipofectamine® RNAiMAX transfection reagent (Invitrogen). Upon 72 h of transfection, SH-SY5Y cells were treated with 25 $\mu$M FCCP. After 6 h of FCCP treatment, dead cells were stained with BOBO™-3 Iodide (Thermo Fisher Scientific) and counted with Operetta microplate imaging reader, the Acapella and Columbus image data processing system, and software (PerkinElmer).

## Mature miRNA quantitative assay

For mature quantitative detection of miR-181a and miR-181b, we used hydrolysis probes (TaqMan, Applied Biosystem). The cDNAs for mature miRNA analysis were generated using TaqMan MicroRNA Reverse Transcription Kit with miRNA-specific primers, according to manufacturer's instructions. The quantification data, obtained in TaqMan-PCRs on TaqMan-cDNAs from different treatments, are expressed in terms of cycle thresholds (*Ct*). A TaqMan probe for the RNA *sno234* was used as endogenous control for the experiments. The Ct values were analyzed as described in the "Quantitative Real-Time PCR" section. Each plate was performed in duplicate, and all results are shown as means $\pm$ SEM of three independent biological replicates.

## RNA *in situ* hybridization (ISH)

Mouse eyes were fixed overnight in 4% PFA, cryoprotected with 30% sucrose, and embedded in OCT. Twenty-micrometer cryosections were treated with 5 $\mu$g/ml proteinase K for 15 min. After washes with 2 mg/ml glycine and post-fixation with 4% PFA/0.2% glutaraldehyde, sections were prehybridized with 50% formamide, 5× sodium saline citrate buffer (SSC) and citric acid to pH 6.1% sodium dodecyl sulfate (SDS), 500 $\mu$g/ml yeast tRNA, and 50 $\mu$g/ml heparin. The miRCURY detection miR-181a and miR-181b locked nucleic acid probes (Exiqon) were used at a final concentration of 30 nM. Probe hybridization was performed overnight at 42°C. Hybridized sections were washed with 50% formamide, 2 × SSC at the hybridization temperature. Sections were blocked for 1 h in MABT (100 mM maleic acid, 150 mM NaCl, and 0.1% Tween-20) containing 1% blocking reagent (Roche) and 10% sheep serum and incubated with alkaline phosphatase (AP)-labeled anti-digoxigenin antibody (1:2,000; Roche) in MABT/1% blocking reagent overnight at 4°C. Sections were stained with NBT-BCIP (Roche) and photographed under a Leica DM5000 microscope.

## Quantitative mitochondrial DNA content analysis

SYBR Green qPCR was performed using primers for the *MT*-CO1 gene (mtDNA) and the RNaseP (nuclear gene reference), as described (Viscomi *et al*, 2009). mtDNA relative copy number was calculated from threshold cycle value ($\Delta$Ct), and mtDNA copy number/cell was calculated as $2 \times 2^{-\Delta Ct}$ to account for the two copies of RNaseP in each nucleus.

## Protein isolation and Western blotting (WB)

Tissue and cells samples for total protein extraction were homogenized in RIPA buffer with Protease Inhibitor Cocktail Tablet (Roche). Protein extract concentrations were determined using the Bio-Rad protein assay (Bio-Rad). A total of 30 $\mu$g protein from each sample was loaded in SDS–PAGE. Mitochondrial and cytosolic extracts were obtained from whole eyes after lens removal, as described (Comitato *et al*, 2014). Thirty micrograms of proteins from mitochondrial and cytosolic fractions from each sample was loaded in SDS–PAGE. For WB, gels were electroblotted onto PVDF filters (Millipore) and sequentially immunostained with the following primary antibodies: anti-p62 (Abnova, clone 2C11, 1:1,000); anti-LC3B (Novus, 1:500); anti-Mfn2 (Abcam, ab56889, 1:1,000); anti-citrate synthase (Abcam, ab9660, 1:1,000); anti-p115 (Santa Cruz, sc-48363, 1:3,000); anti-Tim23 (Santa Cruz, sc-13298, 1:250); anti-Ndufb11 (Proteintech, 1:1,000); anti-Ndufb8 (Abcam, Total OXPHOS Rodent WB Antibody Cocktail ab110413, 1:250); anti-CoxIV (Cell Signaling 4844, 1:1,000); anti-Parkin (Cell Signaling 4211, 1:500); anti-Gapdh (Santa Cruz sc-32233, 1:3,000).

Proteins of interest were detected with horseradish peroxidase-conjugated goat anti-mouse or anti-rabbit IgG antibody (1:3,000, GE Healthcare) visualized with the Luminata Crescendo substrate (Millipore) or the Super Signal West Femto substrate (Thermo Scientific), according to manufacturer's protocol. WB images were acquired using the ChemiDoc-It imaging system (UVP), and band intensity was calculated using the ImageJ software. The signals for each protein staining were quantified and then normalized for GAPDH, p115 or Ndufb8, CS (mitochondrial fractions) in the same sample (internal normalization). These normalized values were then compared to the values in the control sample. Only bands on the same blot were compared. The average of the normalized values from three different biological replicates is reported as the relative fold change.

## Medakafish and morpholino

Medakafish were staged as described (Iwamatsu, 2004). Design, specificity, and inhibitory efficiency of MOs were previously described (Indrieri *et al*, 2012, 2013; Carrella *et al*, 2015). *hccs*-MO, *cox7B*-MO, miR-181a-MO, and miR-181b-MOs were injected into fertilized one-cell medaka embryos at a concentration of 0.3, 0.1, 0.075, and 0.075 mM, respectively.

## TUNEL assay

Whole-mount TUNEL assay was performed on stage(st)30 medakafish embryos using *In Situ* Cell Death Detection Kit (Roche) as described (Indrieri *et al*, 2013). After staining, embryos were embedded in a mix of BSA/Gelatine and vibratome-sectioned. Sections were analyzed with a Leica DM-6000 microscope, and TUNEL-positive cells were manually counted in the entire retina.

## Caspase assays

Twenty embryos for sample were dechorionated at st32 and frozen in liquid nitrogen for at least 24 h. Caspase assays were performed on protein lysates using caspase-3/7- and caspase-9-GLO luciferase reagent (Promega) as previously reported (Indrieri *et al*, 2013). The emitted luminescence signals were normalized for the protein concentration of each sample.

## Drug treatments in medakafish

st24 embryos were dechorionated and incubated for 24 h in 50 nM Bafilomycin A (Sigma), 25 µM PD98059 (Cell Signaling), or 6 µM HA14-1 (Sigma). Drugs were diluted in 3% DMSO/embryo medium. Medium was refreshed twice a day. Control embryos were grown in 3% DMSO/embryo medium.

## Rotenone injections

Mice were injected intravitreally with 2 µl of rotenone (Sigma) 5 mM, as reported (Heitz *et al*, 2012). For histological analysis of RGCs, one murine eye was injected with rotenone and the contralateral eye with DMSO (internal control). For optokinetic tests, mice were bilaterally injected with rotenone 5 mM or DMSO.

## Optokinetic tracking

Visual acuity of rotenone-injected mice was assessed by using the optomotor system (OptoMotry; Cerebral Mechanics) as described (Chadderton *et al*, 2013).

## Immunofluorescence and immunohistochemistry analysis

For immunofluorescence analysis, mouse eyes were fixed in 4% PFA, cryoprotected with 30% sucrose, embedded in OCT, and cryosectioned. Sections were permeabilized by boiling in sodium citrate buffer or by incubation with 1% NP-40 for 15 min for cone-arrestin immunostaining. The following primary antibodies were incubated overnight: anti-Rhodopsin (Abcam, ab3267, 1:5,000); anti-cone arrestin (Millipore, AB15282, 1:1,000); anti-Pax6 (Covance, 42-6600, 1:250); anti-GS6 (Millipore, MAB 302, 1:100); anti-Syntaxin (Sigma, S1172, 1:100); anti-NeuN (Millipore, clone A60, 1:400); and anti-CoxIV (Cell Signaling 4844, 1:400). Sections were then incubated with the Alexa Fluor secondary antibodies (1:1,000; Invitrogen). Sections were counterstained with DAPI (Vector Laboratories). RGCs stained using anti-NeuN antibody were counted on eight different slides for each eye in the areas surrounding the optic nerve. Sections were photographed under a LSM710 Zeiss confocal microscopy.

## Electroretinogram (ERG)

Mice were dark-reared for 3 h and anesthetized. Flash electroretinograms (ERGs) were evoked by 10-ms light flashes generated through a Ganzfeld stimulator (CSO, Costruzione Strumenti Oftalmici, Florence, Italy) and registered as previously described (Surace *et al*, 2005; Botta *et al*, 2016). ERGs and b-wave thresholds were assessed using the following protocol. Eyes were stimulated with light flashes increasing from $-5.2$ to $+1.3$ log cd*s/m$^2$ (which correspond to $1 \times 10^{-5.2}$ to $20.0$ cd*s/m$^2$) in scotopic conditions. For ERG analysis in scotopic conditions, the responses evoked by 11 stimuli (from $-5.2$ to $+1.3$ log cd*s/m$^2$) with an interval of 0.6 log unit were considered. To minimize the noise, three ERG responses were averaged at each 0.6 log unit stimulus from $-5.2$ to $0.0$ log cd*s/m$^2$, while one ERG response was considered for higher stimuli (from $0.0$ to $+1.3$ log cd*s/m$^2$). a- and b-wave amplitudes recorded in scotopic conditions were plotted as a function of increasing light intensity (from $-4$ to $+1.3$ log cd*s/m$^2$). The photopic ERG was recorded after the scotopic session by stimulating the eye with ten 10-ms flashes of $20.0$ cd*s/m$^2$ over a constant background illumination of 50 cd/m$^2$.

## Electron microscopy and mitochondrial count

Mice were deeply anesthetized and perfused with 1% glutaraldehyde and 2% PFA in 200 mM HEPES buffer, pH 7.3, through the heart. Eyes were removed and left for 1 h in the same fixative solution. Specimens of retina were post-fixed in 1% osmium tetroxide, dehydrated, and embedded in epoxy resin. Retina samples were cut on ultra-microtome Leica EM UC7 and collected on the single-slot oval grids and analyzed with FEI electron microscope. Mitochondrial number/cell was determined using FEI software, by counting the number of mitochondria in RGCs ($N \geq 10$ cells for each animal).

## Mitochondrial enzymatic activity measurements

Mitochondria were isolated from the two eyes of one mouse as described (Comitato *et al*, 2014). The mitochondrial pellets were kept frozen at $-80°C$ until the measurements were performed. Each pellet was resuspended in washing buffer (250 mM sucrose, 20 mM HEPES-KOH, pH 7.4, 1 mM EDTA), and one cycle of freeze (in liquid N$_2$) and thawing (at 37°C) was performed. The activities of MRC complex I (NADH-CoQ$_1$ oxidoreductase), complex II (succinate-CoQ$_1$-DCIP oxidoreductase), and complex IV (cytochrome *c* oxidase) and of citrate synthase were measured spectrophotometrically in a 96-well plate reader, as described (Tiranti *et al*, 1995; Kirby *et al*, 2007).

**The paper explained**

**Problem**

Mitochondrial dysfunction underlies the pathogenesis of a variety of human neurodegenerative disorders, including mitochondrial diseases (MDs), a heterogeneous group of devastating and often fatal disorders due to defective oxidative phosphorylation. The complexity and genetic heterogeneity of these disorders have so far prevented the development of effective therapeutic strategies. Therefore, the establishment of gene-independent therapeutic strategies for MDs characterized by neuronal degeneration represents a priority.

**Results**

Here, we identify two microRNAs, miR-181a and miR-181b, as new global regulators of mitochondrial turnover in the central nervous system. We demonstrate that their modulation ameliorates the disease phenotype in *in vivo* models of primary mitochondrial diseases, namely microphthalmia with linear skin lesions and Leber's hereditary optic neuropathy.

**Impact**

Altogether, our findings indicate that these microRNAs may represent novel gene-independent therapeutic targets for mitochondrial diseases and other neurodegenerative disorder associated with mitochondrial dysfunction for which an effective therapy is not available to date.

## Statistical analysis

Sample sizes were estimated on the basis of power analysis assuming significance when $P < 0.05$ and power $\geq 0.8$, and on the basis of our previous experience. In general, for each experiment we used $\geq 3$ animals per genotype, in order to obtain statistically suitable values. Animals/samples were randomly allocated in the different groups based on the appropriate genotype/conditions/treatments. Data were excluded from the analysis if an animal died during experiments. Because of the obvious clinical and morphological phenotypes of most of the animals, the study was not based on blinding of the investigators. The number of experimental replicates is indicated in each figure legend. In all experiments, significance of differences between groups was evaluated by one-way or two-way ANOVA with *post hoc* Tukey's analysis, one-tailed or two-tailed Student's *t*-test, and analysis of deviance for generalized linear model or analysis of deviance for negative binomial generalized linear model as reported in figure legends. $P < 0.05$ was considered significant. Quantitative data are presented as the mean $\pm$ SEM (standard error of the mean) of at least three independent experiments.

**Expanded View** for this article is available online.

## Acknowledgements

We thank Drs. Auricchio, Casari, Settembre, and Morleo for discussion and critical reading of the manuscript. We thank the TIGEM Medaka, Microscopy, and Bioinformatics Cores, and Drs. Nusco, Crispino, and Salierno for technical assistance. We thank the Italian Fondazione Telethon (grant no. TGM16YGM02 to S. Ban, the Fondazione Roma (grant no. RP-201300000009 to S. Ban)) and the AFM-Telethon (grant no. 20685 to B.F.). A.I. received an Umberto Veronesi Fellowship. This research was carried out in the frame of Programme STAR, financially supported by UniNA and Compagnia di San Paolo (Bando STAR, 16-CSP-UNINA-048, to A.I).

## Author contributions

AI and SC conceived the study, designed the experiments, analyzed the data, and wrote the manuscript. AI, SC, AR, AS, SBar, MP, YE, FMG, LC, and RT carried out experimental work. EM and EMS performed intravitreal injection and visual tests in mice. EF-V and MZ contributed to the characterization of *Ndufs4* mouse model. EDL and MZ provided expertise and feedback. JH-M, AW, and RAF provided critical reagents for studies in the mouse. SBan and BF conceived the study, wrote the manuscript, and provided funding. All authors discussed the results and had the opportunity to comment on the manuscript.

## Conflict of interest

The authors declare that they have no conflict of interest.

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
