## [Review Process File · EMBO Molecular Medicine]

miR-181a/b downregulation exerts a protective action on Mitochondrial Disease models

Alessia Indrieri, Sabrina Carrella, Alessia Romano, Alessandra Spaziano, Elena Marrocco, Erika Fernandez-Vizarra, Sara Barbato, Mariateresa Pizzo, Yulia Ezhova, Francesca M. Golia, Ludovica Ciampi, Roberta Tammaro, Jorge Henao-Mejia, Adam Williams, Richard A. Flavell, Elvira De Leonibus, Massimo Zeviani, Enrico M. Surace, Sandro Banfi, Brunella Franco

Review timeline:

Submission date:	29 November 2017
Editorial Decision:	6 December 2017
Authors' appeal:	18 December 2017
Editorial Decision:	5 February 2018
Resubmission:	14 December 2018
Editorial Decision:	22 January 2019
Revision received:	13 February 2019
Editorial Decision:	7 March 2019
Revision received:	14 March 2019
Accepted:	18 March 2019

Editor: Céline Carret

Transaction Report:

1st Editorial Decision

6 December 2017

Thank you for the submission of your manuscript "miR-181a/b Downregulation Protects From Mitochondria-associated Neurodegeneration". I have now had the opportunity to carefully read your paper and the related literature and I have also discussed it with my colleagues. I am afraid that we concluded that the manuscript is not well suited for publication in EMBO Molecular Medicine and have therefore decided not to proceed with peer review.

We appreciate that you identify here miR-181a/b as a regulator of mitochondria biogenesis through autophagy in the CNS and that genetic down regulation of it protects from mitochondria dysfunction in several animal models. While the insight will certainly be relevant to the more immediate community, I am afraid that we are not certain that the conceptual advance be sufficiently striking (given what's known on miR-181 already), nor the translational implications be strong enough, for the paper to be further considered.

I am sorry to have to disappoint you on this occasion.

Please rest assured that this is not a judgment of the quality or interest of your work but a decision based on appropriateness for EMBO Molecular Medicine.

Authors' appeal

18 December 2017

Thank you for the rapid assessment of our manuscript. We respectfully disagree with your conclusion that the conceptual advance of our paper is not sufficient to warrant consideration in EMBO Mol Med given what's known already on miR-181. To the best of our knowledge what is known already on miR-181 includes:

- a role in autophagy and mitophagy as assessed exclusively by in vitro experiments in cell lines (PMID: 23322078 and PMID: 27281615)
 - the targeting of Bcl-2 family members thus influencing cell death, oxidative stress and mitochondrial membrane potentials in astrocytes (PMID:21958558). However, the claim of the latter publication that miR181a influences general mitochondrial function in astrocytes is not supported by experimental data

Conversely, we now demonstrate that:

- miR-181a/b activity is crucial in the global regulation of mitochondrial turnover in the central nervous system. This role is corroborated by the completely new finding that these two miRNAs control mitochondrial biogenesis and by the evidence that they modulate autophagy/mitophagy in vivo. To the best of our knowledge, the simultaneous control on mitochondrial biogenesis and clearance constitutes an unprecedented example for a single gene in the central nervous system.
 - miR181a/b downregulation exerts, in a gene-independent manner, a significant neuroprotective effect in several in vivo models for three different mitochondrial-mediated neurodegenerative diseases, namely Microphthalmia with Linear Skin Lesions, Leber hereditary optic neuropathy and Parkinson's disease.

Based on these considerations we would like to draw your attention on the fact that the lack of novelty does not seem to be a valid issue for considering our manuscript not suitable for further evaluation by EMBO Mol Med. Therefore we wonder whether you could reconsider your decision and we look forward to hearing from you.

2nd Editorial Decision

5 February 2018

Thank you for the submission of your manuscript "miR-181a/b Downregulation Protects From Mitochondria-associated Neurodegeneration". We have now heard back from the three referees whom we asked to evaluate your manuscript.

As you will see from the set of comments pasted below, the referees acknowledge the potential interest of the study. Unfortunately, all three have very serious and very overlapping concerns that I am afraid to say, preclude further processing of your article for EMBO Molecular Medicine.

I apologize that we were unable to return to you with this answer any sooner, and am sorry to disappoint you on this occasion. I hope, however, that the referee comments will be helpful in your continued work in this area.

***** Reviewer's comments *****

Referee #1 (Comments on Novelty/Model System for Author):

The models used are well known to the authors or to the mitochondrial field, however they are not really reflecting the classical mitochondrial disease (MD). MLSL is a very rare disorder characterized by a phenotype that is rarely encountered in MDs. On the other hand, rotenone and 6OHDA are toxins that induce rapid cell death.

Referee #1 (Remarks for Author):

This interesting study explores the possible therapeutic implications of miR-181a/b in protection from cell death in genetic or pharmacological models of neurodegeneration linked to mitochondria dysfunction. The paper makes substantially three claims (see also Abstract):

- 1) miR-181a/b controls genes involved in mitochondrial biogenesis and function
- 2) miR181a/b regulates pathways involved in mitochondrial turnover
- 3) Downregulation of miR181a/b is an efficient therapeutic strategy in various models of neurodegeneration linked to mitochondrial dysfunction.

I was impressed by the extent of the phenotype rescues that the authors show. These results are solid, supported by the data, and very promising. They suggest the need to consider these miRNAs

as valid therapeutic targets. However, the data in this manuscript are not totally convincing as to the mechanism by which the miRNAs act, in other words which are the targets of the miRNAs that are responsible for the effects observed. Also, the choice of the models reflects severe phenotypes characterized by cell death, and not typical mitochondrial diseases.

Specific comments:

1) The western blots in Figure 1E are of poor technical quality and I can hardly see the effects in the blot shown. The parkin western blot is also not very convincing. Why the pattern of parkin is different in WT and KO? The question that I was left with after reading this manuscript is in fact: how many target of miR181a/b are indeed encoding mitochondrial proteins? And, is so, is this really physiologically relevant? The authors should provide more data showing increased mitochondrial mass in the tissues of miR181a/b knock-outs (for instance, mitochondria mass can be estimated by immunofluorescence, electron microscopy, and by performing more and better western blots using several markers for mitochondria, including outer membrane proteins). It would be important to show that this happens also in at least one of the model systems that they use. They could also test if the respiratory function increases in absence of the miRNAs in different tissues (especially brain).

2) The involvement of parkin and therefore mitophagy in the observed effects is also poorly demonstrated. First of all, the implication of parkin in mitophagy *in vivo* in the mouse has not been demonstrated conclusively yet. Secondly, the only experiment that the authors show to implicate mitophagy in the phenotype rescue is the abolishment of the effect in presence of bafilomycin. This is a drug that can have multiple effects. Genetic strategies would be more convincing, for instance downregulation of Atg5.

3) The authors use initial two medaka models of microphthalmia with linear skin lesions, a very rare condition caused by mutations in different genes encoding mitochondrial proteins. They then extend their findings to models of neurodegeneration caused by mitochondrial toxins both in mouse and medaka. All these models have in common mitochondrial dysfunction, but also prominent cell death, and do not really reflect the classical mitochondrial diseases. Despite these outstanding results, I am not completely convinced from the data shown here that the reason for the protection from neurodegeneration is to be attributed to the function of miR-181a/b on mitochondrial biogenesis/turnover. The authors themselves very honestly discuss their finding in light on all previous knowledge on the target of miR-181a/b. Among these there are for instance anti-apoptotic genes, so it is possible that what they are looking at here is a delay of apoptosis. This in fact seems to be the case in the Hccs model. The paper would be strengthened by using additional models of mitochondrial dysfunction characterized by respiratory deficiency and only later by cell death. In such a model, they could explore if respiratory deficiency is delayed or rescued even before cell death occurs.

Referee #2 (Comments on Novelty/Model System for Author):

Some of the models utilized do not necessarily reflect primary mitochondrial diseases.

Referee #2 (Remarks for Author):

The manuscript by Carrella and colleagues describes the protective effects of miR-181a/b downregulation regulation in a variety of models of cell stress and toxicity, *in vitro* and *in vivo*. They demonstrate that modulation of miR-181a/b results in upregulation of the transcripts of a number of mitochondrial proteins in SH-SY5Y cultured neuroblastoma cells. They use miR-181a/b KO mice eyes to confirm the upregulation of mitochondrial genes *in vivo*. They also suggest that miR-181a/b induces autophagy and mitophagy. In fish models of a neurodevelopmental disorder, MLS syndrome, they show that miR-181a/b silencing improves neuronal loss in the brain and in the eye, in an autophagy dependent manner. Interestingly, inhibition of Bcl-2 activity prevents the protection conferred by miR-181a/b silencing only in one of the two genetic models tested. Then, they move to a chemical model of optic neuropathy to show that miR-181a/b KO mice are protected against rotenone induced RGC degeneration. Lastly, they look at the protective effects of miR-181a/b KO in fish and in mice treated with 6-OHDA to induce dopaminergic neuron death.

Overall, the results in the many systems tested clearly show that miR-181a/b downregulation is protective against a number of pro-death insults. However, as designed, this manuscript does not provide novel and convincing mechanistic insights, beyond the known effects of miR-181a/b on apoptosis. More work is needed to dissect the nature of the protection in relationship to mitochondrial dysfunction in the context of mitochondrial diseases. Therefore, based on the evidence presented here, the conclusions are overstated.

Specific points:

- 1) The models utilized in this work are extremely heterogeneous. This is not necessarily a weakness, but the problem is that the models that the authors define as mitochondrial diseases do not convincingly recapitulate well any of the known primary mitochondrial disorders.
- 2) The majority of the protection effects observed can be attributed to the known modulatory effect of miR-181a/b on apoptosis. The exception appears to be the *cox7b*-MO fish model of MLS syndrome, as it was unaffected by HA14-1. This is intriguing and would be worth a mechanistic follow up, for example by excluding other apoptotic pathways involved.
- 3) Overall, with the exception of the work done in cultured SH-SY5Y, there is very little characterization of the mechanisms taking place in each of the model investigated. The work is too descriptive to be able to come to firm conclusions on protective effects of miR-181a/b besides general anti-apoptotic mechanisms.
- 4) The mitophagy pathway is also of potential interest, but there are no indications on whether it would be subject to an upstream regulation or be modified secondarily, as a consequence of mitochondrial outer membrane permeabilization by pro-apoptotic proteins.
- 5) The chemical toxins utilized in vitro (extremely high doses of FCCP) or in vivo (rotenone, 6-OHDA) are paradigms of acute stress and do not necessarily mimic any real mitochondrial disease. It would be much more interesting to investigate the effects of miR-181a/b KO or silencing in better mouse models of primary mitochondrial dysfunction, such as the *NDUFS4* KO mouse or the *ND6* mutant mice.
- 6) The effects on mitochondrial biogenesis and function are only investigated in SH-SY5Y cells and not in the disease models in which they see protection.
- 7) In some cases, the results presented are unexplained. Why, for example, would miR-181a/b KO prevent complex I inhibition by rotenone, and why do the authors expect that rotenone should affect CoxIV immunofluorescence?
- 8) No systematic and exhaustive analyses on the effect of miR-181a/b modulation on mitochondrial activity, autophagy, mitophagy, and apoptosis have been performed in any of the toxic/disease models. The various studies are too piecemeal to be able to interpret the results and draw convincing mechanistic conclusions.

Referee #3 (Comments on Novelty/Model System for Author):

The model systems used are INADEQUATE due to the fact that several models are used (cell lines, medakafish, mouse) to test different aspects but no rationale is presented for using these model systems.

Referee #3 (Remarks for Author):

Carrella and co-workers describe within this manuscript that the downregulation of the miR-181a/b protects neurons from a mitochondrial-associated degeneration. For this, the authors use several model system with different neuropathology's, but the common denominator of a possible mitochondrial dysfunction. Even though this is a rather interesting subject and that has some impact within the scientific community, the work presented in the manuscript requires further studies to validate the major claims stated throughout the text.

1. Using SH-SY5Y cells, the authors validate newly identified miR-181a/b targets and conclude that based on these results that the downregulation if this miRNA stimulates mitochondrial biogenesis. However, no assessment of mitochondrial biogenesis is directly assessed in this cell-based model. Authors should experimentally determine this mitochondrial properties to support such a statement, for example by determining PGC-1 α and TFAM levels. Additionally, the authors also use the miR-181a/b KO mouse where, among other experiments, measured mtDNA copy number in the mouse eye as well as protein levels of LC3-II and p62. As mentioned above, the authors could have profited from this mouse model and determined mitochondrial properties, such as biogenesis and

mitochondrial number and clearance, in order to further support their claims. Furthermore, assuming that mitophagy is occurring due to the increased levels of LC3-II and p62 is rather indirect. And here again the authors could directly look at mitochondria morphology and determine if mitophagy is occurring.

2. Authors use three types of mitochondrial dysfunction disorders to further validate the role of miR-181a/b in mitochondrial biogenesis. They analysed MLS syndrome, LHON syndrome and PD in a morpholino-based medaka fish approach and a rotenone and 6-OHDA induced toxin mouse models, respectively. The rationale for using different model systems is not explained, and in neither of the models is direct mitochondrial morphology assessed. In most approaches used, no data validating that the applied conditions were working properly were presented (for example: effects of rotenone on Complex I activity, effects of 6-OHDA...)

3. In the specific case of the PD model and its connection to the miR-181a/b, the authors should consider integrating the publication from Liu et al. 2017, that addresses the importance of exactly the same miRNA in this pathology.

Overall, and based on the results presented within this manuscript, the conclusion that this work "provide a proof-of-principle that inactivation of miR-181a/b protects from mitochondria mediated neurodegeneration" is somewhat of an overstatement, especially when no direct assessment of mitochondrial morphology or turnover was performed.

Resubmission

14 December 2018

On behalf of all co-authors please find enclosed our manuscript entitled: "miR-181a/b downregulation exerts a protective function on mitochondrial disease models", by Indrieri, Carrella et al., for submission as a Research Article to EMBO Molecular Medicine.

In this manuscript, we uncover the microRNAs miR-181a and miR-181b (miR-181a/b) as novel therapeutic targets for Mitochondrial Diseases characterized by neuronal degeneration. These two microRNAs were previously reported to target *in vitro* genes involved in mitochondria-dependent cell death and autophagy. We now find that miR-181a/b are involved in the control of mitochondrial biogenesis and function *in vivo*, in the central nervous system. To the best of our knowledge, the simultaneous control of mitochondrial biogenesis and clearance constitutes an unprecedented example for noncoding RNAs, and highlights miR-181a/b as master regulators of mitochondrial homeostasis in the central nervous system.

We then demonstrated that downregulation of miR-181a/b exerts a protective effect *in vivo* in models of Mitochondrial diseases, namely Microphthalmia with linear skin lesions and Leber Hereditary Optic Neuropathy. In particular, the analysis of a genetic model for a typical mitochondrial disorder, i.e., the *Ndufs4* KO mouse, allowed us to directly assess the consequence of miR-181a/b inactivation on mitochondrial biogenesis/clearance, morphology and function. Our findings suggest that the strong neuroprotective effect of miR-181a/b downregulation is mediated by the balanced control of mitophagy and mitochondrial biogenesis resulting in the global regulation of mitochondrial turnover.

Importantly, our results also indicate that the neuroprotective effect of miR-181a/b downregulation is independent on the specific genetic defect underlying the neurodegeneration. As a result, this approach holds promise for a potential therapeutic application to Mitochondrial Diseases and other neurodegenerative diseases associated with mitochondrial dysfunction for which an effective therapy is not available to date.

We believe that biological relevance and significance of the results presented here are of interest for the broad readership of EMBO Molecular Medicine and therefore we hope you find the manuscript suitable for publication in this journal.

Please note that this report is a profoundly modified version of a previously submitted manuscript (EMM-2017-08734-V2-Q) handled by Dr. Celine Carret.

Thank you for the submission of your manuscript to EMBO Molecular Medicine. We have now heard back from two of the initial referees who reviewed your article the 1st time. Although the referees find the study to be improved particularly with your choice of models showing neuroprotection in the retina, the mechanism by which this occurs remains unclear and we would like to encourage you to address this as convincingly as you can, by looking into the effects of the known anti-apoptotic role of silencing miR-181a/b.

Please note that this is an exceptional scenario as EMBO Molecular Medicine policy normally allows only one round of revision.

Should the paper be further considered, please do pay extra attention to our guidelines to format your paper as needed for publication.

I look forward to seeing a revised form of your manuscript as soon as possible.

***** Reviewer's comments *****

Referee #1 (Remarks for Author):

This interesting study investigates the possible therapeutic role of miR-181a/b downregulation in several genetic or pharmacological models of retinal degeneration caused by respiratory deficiency. This revised version convincingly shows the protective role of targeting miR-181a/b in several models, including rare forms of MLS in medaka, *Ndusf4* KO mice and rotenone-induced retinal ganglion cell degeneration. These are important results that indicate miR-181a/b as novel therapeutic target potentially for a wide class of mitochondrial diseases. In particular, the new data in *Ndusf4* model really add an important piece to this story, showing rescue in a paradigmatic mitochondrial disease, and demonstrating in vivo the increase in mitochondrial number, the improvement of mitochondrial morphology, and in respiratory function upon loss of the miRNA. Although the mechanism by which this rescue occurs is still only partially developed, I believe that these data are important and worth being reported.

Specific comments:

- 1) The authors propose as mechanism of action a concomitant increase in mitochondrial biogenesis and mitophagy. Although this concept is attractive, the data in support of it are still a bit sparse and not entirely convincing. One problem is that the authors look at a few very specific gene targets of miR-181a/b (which are nicely validated). The results are not always consistent in the different models (however, this can be expected). It is well possible that other target genes contribute to the rescue. A few questions remain: Is NRF1 also increased at the protein level? If so, wouldn't one expect to see an upregulation of several genes controlled by NRF1? It seems from the data presented that only a few mitochondrial proteins increase in abundance. RNA-seq or proteomics may help having a broad picture of how the mitochondrial proteome is affected, but it may be outside the scope of the present work. If NRF1 is the main driver of the rescue, as the authors propose, is it possible to impair the rescue by downregulating NRF1, at least in cells treated with FCCP?
- 2) Increased recruitment of p62 and parkin does not necessarily mean that productive mitophagy occurs. Mitophagy could be for example blocked at a late step. It may be worth to at least discuss it. Furthermore, there are no signs of mitophagy in Figure 6B. Parkin overexpression could be beneficial in several other ways than pushing mitophagy. This should be discussed.
- 3) Concerning the experiment with the inhibitors: what is the effect of the inhibitors on the mutants?
- 4) Figure 4D: Cox4 staining quality hardly shows increased mitochondrial number or the network integrity. The authors should use super-resolution microscopy or provide better pictures (or enlargements).
- 5) Do *Ndusf4*^{-/-} miR-181a/b^{-/-} mice display prolonged survival and protection from neurodegeneration? This would show that the mechanism is not restricted to the retina. Independent on the results, if the authors possess these data, I would encourage them to include them.
- 6) Fig. 5F and G: t-tests between WT and double KO should also be included.

- 7) Methods: Specification about the genetic background of the mice and how crossing were made should be included. Are controls littermates? This should be specified, as well as the housing conditions and the diet.
- 8) Methods: More detail on the quantification of electron microscopy should be included (how many sections, how many cells from how many animals).

Referee #2 (Remarks for Author):

This revised manuscript by Indrieri, Carrella, and colleagues investigates the neuroprotective effects of miR-181a/b downregulation. In this revision, they have responded to previous comments by removing the 6-OHDA toxicity models and adding the NDUFS4 KO mouse model of Leigh syndrome. Also in this mouse model of mitochondrial disease, they demonstrate that miR-181a/b KO alleviates the eye phenotype. Therefore, the protective effects of miR-181a/b downregulation against a multiplicity of mitochondrial stressors in RGC is convincingly demonstrated. They now show interesting effects of miR-181a/b downregulation on mitochondrial gene expression and mitochondrial content, suggesting that these miRs may affect mitochondrial biogenesis and, possibly, mitochondrial turnover. However, it remains unclear if the neuroprotection is due to mechanisms other than the known effects of miR-181a/b on apoptosis.

- 1) It would be important to know if the transcriptional effects of miR-181a/b downregulation are selective to the retinal neurons or if they extend to other neuronal populations, and other cell types.
- 2) The protection at 6 hours after mitochondrial depolarization in SH-SY5Y neuron-like cells by FCCP is likely associated with anti-apoptotic effects, rather than increased mitochondrial biogenesis or increased mitophagy.
- 3) In figure 3, there is no control + bafilomycin treatment group as a reference. Therefore, it is difficult to assess if the drug has an effect independent of the silencing of complex III and complex IV genes.
- 4) The different responses to HA14-1 in hccs-MO/miR-181a/b-MO embryos, and in the cox7b-MO/miR-181a/b-MO embryos remain unexplained.
- 5) There is no reason why NADH dehydrogenase activity should be spared by miR-181a/b downregulation in eyes treated with rotenone, except prevention of cell loss.
- 6) Which are the consequences of ablating miR-181a/b on the survival and the neuronal death of NDUFS4 KO mice, beyond RGC?
- 7) The values of complex I activity of *Ndufs4*^{-/-}/miR-181a/b-1^{-/-} mice compared with the *Ndufs4*^{-/-} mice should also be shown to appreciate the magnitude of the effect. Again, it is unclear why miR-181a/b downregulation would increase complex I activity in the NDUFS4 KO.
- 8) The protective effects of rapamycin in mouse models of mitochondrial disease was not attributed to autophagy but to metabolic shifts.

1st Revision - authors' response

13 February 2019

Referee #1 (Remarks for Author):

This interesting study investigates the possible therapeutic role of miR-181a/b downregulation in several genetic or pharmacological models of retinal degeneration caused by respiratory deficiency. This revised version convincingly shows the protective role of targeting miR-181a/b in several models, including rare forms of MLS in medaka, *Ndufs4* KO mice and rotenone-induced retinal ganglion cell degeneration. These are important results that indicate miR-181a/b as novel therapeutic target potentially for a wide class of mitochondrial diseases. In particular, the new data in *Ndufs4* model really add an important piece to this story, showing rescue in a paradigmatic mitochondrial disease, and demonstrating in vivo the increase in mitochondrial number, the improvement of mitochondrial morphology, and in respiratory function upon loss of the miRNA. Although the mechanism by which this rescue occurs is still only partially developed, I believe that these data are important and worth being reported.

Specific comments:

- 1) The authors propose as mechanism of action a concomitant increase in mitochondrial biogenesis

and mitophagy. Although this concept is attractive, the data in support of it are still a bit sparse and not entirely convincing. One problem is that the authors look at a few very specific gene targets of miR-181a/b (which are nicely validated). The results are not always consistent in the different models (however, this can be expected). It is well possible that other target genes contribute to the rescue. A few questions remain: Is NRF1 also increased at the protein level? If so, wouldn't one expect to see an upregulation of several genes controlled by NRF1? It seems from the data presented that only a few mitochondrial proteins increase in abundance. RNA-seq or proteomics may help having a broad picture of how the mitochondrial proteome is affected, but it may be outside the scope of the present work. If NRF1 is the main driver of the rescue, as the authors propose, is it possible to impair the rescue by downregulating NRF1, at least in cells treated with FCCP?

OUR RESPONSE: As the reviewer suggested it is possible that other pathways besides mitochondrial biogenesis and mitophagy may contribute to the neuroprotective effects achieved upon miR-181a/b downregulation. This concept is extensively discussed in pages 13-16. Nevertheless, we strongly believe that enhanced mitochondrial biogenesis and mitophagy play a major role in the amelioration of the phenotype compared, for example, to the effect of these miRNA on apoptosis for the following reasons. Indeed, we show that miR-181a/b depletion induces rescue of visual function as detected by OKR and ERG analyses at P25 and P30, respectively in *Ndufs4^{-/-}/miR-181a/b-1^{-/-}* mice. Moreover, we showed in RGCs increased number of mitochondria accompanied by notable amelioration of their morphological structure and by increase of complex I, II and IV activities at P30. These data indicate that the amelioration of the phenotype due to miR-181a/b downregulation precedes the effect on RGCs death, which is detectable in *Ndufs4^{-/-}* mice only after P42. Thus, the neuroprotective effect observed cannot be explained through the effect of miR181a/b on apoptosis, at least in this model.

Secondly, we have now tested the effect of *NRF1* downregulation in SH-SY5Y cells treated with FCCP and demonstrated that *NRF1* inhibition significantly abolishes the mir-181a/b-mediated rescue, indicating that indeed, NRF1 is implicated in the neuroprotective effect exerted by miR-181a/b downregulation in this model. These data have been added to Figure EV5C and described in the results section on page 12, lines 14-17.

Finally, we evaluated expression levels of few *Nrf1* target genes in the available samples. **[Figure removed upon author's request]** As shown below, our data show a general trend of upregulation of *Nrf1* target genes in *Ndufs4^{-/-}/miR-181a/b-1^{-/-}* mice compared to *Ndufs4^{-/-}*. However, the number of samples available and analyzed did not allow to reach statistical significance and we do not believe that the data can be shown in the manuscript in the present form; nevertheless, the data are reported below for the reviewer's consideration.

2) Increased recruitment of p62 and parkin does not necessarily mean that productive mitophagy occurs. Mitophagy could be for example blocked at a late step. It may be worth to at least discuss it. Furthermore, there are no signs of mitophagy in Figure 6B.

OUR RESPONSE: Our data indicate that mitophagy does occur. Electron microscopy (EM) analysis showed improved mitochondrial morphology in *Ndufs4^{-/-}/miR-181a/b-1^{-/-}* mice compared to *Ndufs4^{-/-}*. This can be explained by increased mitochondrial turnover. Conversely, a possible mitophagy block at the level of autophagosome recruitment or autophagosome/lysosome fusion should be evident in EM with a consequent accumulation of autophagosomes/autolysosomes containing undigested material. Moreover, in *miR-181a/b-1^{-/-}* retina, concomitantly with p62 and parkin recruitment to mitochondria, we detected an increased autophagic flux that accelerates mitophagy. We tried to block autophagy to better visualize the process *in vivo* by injection of leupeptin but the drug failed to reach retina and brain and thus the experiment was not conclusive. However, to clarify this issue a sentence has been added in the results section, page 12, lines 4-7.

Parkin overexpression could be beneficial in several other ways than pushing mitophagy. This should be discussed.

OUR RESPONSE: We agree with the reviewers and we added a sentence about this issue in the Discussion, page 14, lines 7-10.

3) Concerning the experiment with the inhibitors: what is the effect of the inhibitors on the mutants?

OUR RESPONSE: We believe that the reviewer is referring to the drugs used to inhibit the pathways in the medaka models. Drugs used in the MLS medakafish models were tested only in controls to determine the appropriate dosage and the results are shown in Figure EV4. Our experiment aimed at inhibiting the pathway/s (autophagy, BCL2-mediated cell death and MAPK) eventually responsible for the miR181-mediated amelioration of the phenotype. This effect was evaluated by looking at the worsening of the phenotype after the use of drugs (BafA1, HA-141 and PD98059, respectively) affecting the above-mentioned pathways. Our control of the experiment was to test the effect of the drugs on controls in order to use a concentration that did not cause per se any abnormalities. We did not test the effect of the drugs on *hccs* or *cox7B* morphants since according to our hypothesis and to previous data (Indrieri, Conte et al., 2013, Indrieri, Grimaldi et al., 2016, Indrieri, van Rahden et al., 2012) these should have been deleterious or even lethal in morphants.

4) Figure 4D: Cox4 staining quality hardly shows increased mitochondrial number or the network integrity. The authors should use super-resolution microscopy or provide better pictures (or enlargements).

OUR RESPONSE: As requested we modified the Figure 4D to increase clarity.

5) Do *Ndufs4*^{-/-} *miR-181a/b*^{-/-} mice display prolonged survival and protection from neurodegeneration? This would show that the mechanism is not restricted to the retina. Independent on the results, if the authors possess these data, I would encourage them to include them.

OUR RESPONSE: We have preliminary results pointing to increased survival in *Ndufs4*^{-/-}/*miR-181a/b*^{-/-} mice as shown below for the reviewers' convenience [**Figure removed upon author's request**]. However, we believe that the number of mice analyzed to date is not sufficient to draw convincing conclusions and that this data should not be included in the manuscript. Nevertheless, we believe that the protective effects of miR181a/b downregulation are not restricted to the retina as also shown by the results obtained using the 6-OHDA toxicity models presented in the initial version of the paper that have now been removed from the present version of the manuscript.

6) Fig. 5F and G: t-tests between WT and double KO should also be included.

OUR RESPONSE: As requested we added the statistical significance between WT and double KO in Figure 5F and G.

7) Methods: Specification about the genetic background of the mice and how crossing were made should be included. Are controls littermates? This should be specified, as well as the housing conditions and the diet.

OUR RESPONSE: Specification about the genetic background, crossing, controls littermates, as well as the housing conditions and the diet of mice and fish were added in the Material and methods section, pages 17-18 in the Animal Studies paragraph.

8) Methods: More detail on the quantification of electron microscopy should be included (how many sections, how many cells from how many animals).

OUR RESPONSE: Details on the quantification of electron microscopy were added in the Materials and methods section, page 24, Electron microscopy and mitochondria count paragraph.

Referee #2 (Remarks for Author):

This revised manuscript by Indrieri, Carrella, and colleagues investigates the neuroprotective effects of miR-181a/b downregulation. In this revision, they have responded to previous comments by removing the 6-OHDA toxicity models and adding the NDUFS4 KO mouse model of Leigh syndrome. Also in this mouse model of mitochondrial disease, they demonstrate that miR-181a/b KO alleviates the eye phenotype. Therefore, the protective effects of miR-181a/b downregulation against a multiplicity of mitochondrial stressors in RGC is convincingly demonstrated. They now show interesting effects of miR-181a/b downregulation on mitochondrial gene expression and

mitochondrial content, suggesting that these miRNAs may affect mitochondrial biogenesis and, possibly, mitochondrial turnover. However, it remains unclear if the neuroprotection is due to mechanisms other than the known effects of miR-181a/b on apoptosis.

OUR RESPONSE: The *Ndufs4*KO mouse represents a model of mitochondrial dysfunction, characterized initially by respiratory deficiency and then by cell death at a later stage (Kruse, Watt et al., 2008). In this model it is possible to explore whether respiratory deficiency is delayed or rescued even before RGCs death occurs (P42) (Kruse et al., 2008, Song, Yu et al., 2017, Yu, Song et al., 2015). Interestingly, we now show that miR-181a/b depletion induces rescue of visual function, as detected by OKR and ERG analyses, at P25 and P30, respectively in *Ndufs4^{-/-}/miR-181a/b-1^{-/-}* mice. Moreover, we showed in RGCs increased number of mitochondria accompanied by notable amelioration of their morphological structure and by increase of complex I, II and IV activities at P30. These data indicate that the amelioration of the phenotype due to miR-181a/b downregulation precedes the effect on RGCs death, which is detectable in *Ndufs4^{-/-}* mice only after P42. Thus, the neuroprotective effect observed cannot be explained through the effect of miR-181a/b on apoptosis, at least in this model. We realize that this concept was not clearly explained and we added the necessary experimental design details (stages and so on) in the Results section (page 11, lines 2, 3, 7, 11, 17) and a paragraph in the Discussion (page 16, lines 4-9).

1) It would be important to know if the transcriptional effects of miR-181a/b downregulation are selective to the retinal neurons or if they extend to other neuronal populations, and other cell types.

OUR RESPONSE: Our preliminary data indicate that indeed the transcriptional effects of miR-181a/b downregulation are not restricted to retinal neurons. We have preliminary data on the analysis of miR-181a/b target levels in the Substantia Nigra (SN) and Retinal Pigmented Epithelium (RPE) of miR-181a/b^{-/-} vs WT mice. Although preliminary, these data suggest that miR-181a/b act also in other neuronal populations and in non-neuronal cell types, with slight differences which can be ascribed to differences among tissues. As stated before (Referee #1, point 5) we showed a protective effect of miR-181a/b downregulation in the 6-OHDA toxicity models that have now been removed from the paper and we have preliminary results showing increase of survival in *Ndufs4^{-/-}/miR-181a/b^{-/-}* mice [**Figure removed upon author's request**]. However, we believe that the number of mice analyzed to date is not sufficient and that these data should not be included in this manuscript and should be the object of a different manuscript.

2) The protection at 6 hours after mitochondrial depolarization in SH-SY5Y neuron-like cells by FCCP is likely associated with anti-apoptotic effects, rather than increased mitochondrial biogenesis or increased mitophagy.

OUR RESPONSE: We agree that this model (SH-SY5Y neuron-like cells + FCCP) is not the best to dissect the effect of miR-181a/b downregulation on the mitochondrial phenotype with respect to the known effect of these miRNAs on cell death. However, as stated before (Referee #1, point 1), we tested the effect of *NRF1* downregulation in SH-SY5Y treated with FCCP. Our data show that *NRF1* downregulation significantly abolish the miR-181a/b-mediated rescue, indicating that also in this model NRF1-mediated mitochondrial biogenesis is implicated in the neuroprotective effect exerted by miR-181a/b downregulation. These data have now been added to Figure EV5C and described on page 12, lines 14-17.

3) In figure 3, there is no control + bafilomycin treatment group as a reference. Therefore, it is difficult to assess if the drug has an effect independent of the silencing of complex III and complex IV genes.

OUR RESPONSE: The control + bafilomycin treatment group is reported in Figure EV4 C-F

4) The different responses to HA14-1 in *hccs*-MO/miR-181a/b-MO embryos, and in the *cox7b*-MO/miR-181a/b-MO embryos remain unexplained.

OUR RESPONSE: *Cox7B* is a structural subunit of complex IV that is indispensable for COX assembly, COX activity, and mitochondrial respiration (Indrieri et al., 2012). *Hccs* catalyzes the covalent attachment of heme to both *Cytc* and *Cytc1*. *Cytc1* is an integral component of complex III that transfers electrons to *Cytc*, which, in turn, transfers the electrons from complex III to complex

IV. We showed that Hccs impairment causes increased cell death via a non-canonical apoptosome-independent caspase-9 activation (Indrieri et al., 2013). We also showed that the unconventional activation of caspase-9 occurs in the mitochondria and is triggered by MRC impairment and overproduction of reactive oxygen species (ROS) (Indrieri et al., 2013). Notably, inhibition of MOMP and PTP, either by overexpression of the Bcl-2 family protein, Bcl-xL, or by cyclosporine A (CsA) treatment, counteracted increased cell death and ameliorated the phenotype of an hccs-deficient MLS medaka (Indrieri et al., 2013). Thus, it is not surprising that miR-181a/b-mediated increase of Bcl-2 and Mcl-1 levels is able to rescue the phenotype of hccs morphants and that, HA14-1, a potent inhibitor of Bcl-2 proteins, significantly abolishes the miR-181a/b-mediated amelioration of the MLS phenotype in the hccs-defective embryos.

Although mutations in either *HCCS* or *COX7B* underlie the MLS syndrome, it is plausible that there are differences in the activation of downstream pathways responsible for the phenotype. Bcl2 overexpression could not be sufficient to ameliorate the phenotype of *cox7B* morphants. As also reported in the discussion of the manuscript, these data indicate that a single pathway, e.g., mitochondrial-dependent cell death, and its modulation, may have different effects on highly related forms of MDs, depending on the underlying genetic defect.

5) There is no reason why NADH dehydrogenase activity should be spared by miR-181a/b downregulation in eyes treated with rotenone, except prevention of cell loss.

OUR RESPONSE: We agree with the reviewer that this model is not suitable to distinguish whether miR-181a/b downregulation acts on the mitochondrial phenotype or cell death. This is an acute model in which we analyzed NADH dehydrogenase activity and RGC number at the same time-point (1-week post-injection). To dissect whether the increase of Complex I activity is due to prevention of cell loss rather than to increased number of mitochondria, that in turn prevent cell loss, we employed *Ndufs4*KO mice in which we were able to investigate this aspect as discussed earlier (see general comments).

6) Which are the consequences of ablating miR-181a/b on the survival and the neuronal death of *NDUFS4* KO mice, beyond RGC?

OUR RESPONSE: As also stated above (Referee #1, point 5 and Referee #2, point 1) we have preliminary results showing increased survival in *Ndufs4^{-/-}/miR-181a/b^{-/-}* mice. This finding indicates a more general effect of miR-181a/b downregulation on neuronal survival.

7) The values of complex I activity of *Ndufs4^{-/-}/miR-181a/b-1^{-/-}* mice compared with the *Ndufs4^{-/-}* mice should also be shown to appreciate the magnitude of the effect. Again, it is unclear why miR-181a/b downregulation would increase complex I activity in the *NDUFS4* KO.

OUR RESPONSE: The values of complex I activity of *Ndufs4^{-/-}/miR-181a/b-1^{-/-}* mice compared with the *Ndufs4^{-/-}* are shown in Figure 6E.

As discussed, the increase of complex I activity in *Ndufs4^{-/-}/miR-181a/b-1^{-/-}*, although significant, is still far from compensating the profound complex I deficiency shown by *Ndufs4^{-/-}* mice. We believe that the slight enhancement of complex I activity is compatible with the miR-181a/b-mediated increased of mitochondrial turnover involving degradation of the most dysfunctional mitochondria and enhanced mitochondrial biogenesis (proposed in Figure 7). This translates in the higher activity of complex I and of the other MRC complexes analyzed.

8) The protective effects of rapamycin in mouse models of mitochondrial disease was not attributed to autophagy but to metabolic shifts.

OUR RESPONSE: Rapamycin has been reported to have a beneficial effect in different mouse models of mitochondrial disease (Civiletto, Dogan et al., 2018, Felici, Buonvicino et al., 2017, Johnson, Yanos et al., 2013, Khan, Nikkanen et al., 2017, Siegmund, Yang et al., 2017) although a general consensus on the mechanisms underlying the effects of rapamycin on primary mitochondrial dysfunction is still lacking. Activation of autophagy upon rapamycin treatment has been reported in few models (Civiletto et al., 2018, Johnson et al., 2013, Khan et al., 2017) while others failed to detect it (Felici et al., 2017, Siegmund et al., 2017). However, these data were based on measurement of autophagy markers at steady state, whereas the autophagic flux was not investigated. Notably one of the coauthors of this paper, Prof Zeviani, recently showed that

rapamycin induces amelioration of mitochondrial function and ultrastructure in muscle-specific Cox15 KO mice, indicating efficient clearance of dysfunctional organelles by parallel activation of both autophagic flux and lysosomal biogenesis in skeletal muscle (Civiletto et al., 2018). We agree that other mechanisms can be involved in mediating rapamycin effects and thus rephrased our sentence in the Discussion (last 3 lines page 13).

References cited in the point by point response to reviewers

- Civiletto G, Dogan SA, Cerutti R, Fagiolari G, Moggio M, Lamperti C, Beninca C, Viscomi C, Zeviani M (2018) Rapamycin rescues mitochondrial myopathy via coordinated activation of autophagy and lysosomal biogenesis. *EMBO Mol Med* 10
- Felici R, Buonvicino D, Muzzi M, Cavone L, Guasti D, Lapucci A, Pratesi S, De Cesaris F, Luceri F, Chiarugi A (2017) Post onset, oral rapamycin treatment delays development of mitochondrial encephalopathy only at supramaximal doses. *Neuropharmacology* 117: 74-84
- Indrieri A, Conte I, Chesi G, Romano A, Quartararo J, Tate R, Ghezzi D, Zeviani M, Goffrini P, Ferrero I, Bovolenta P, Franco B (2013) The impairment of HCCS leads to MLS syndrome by activating a non-canonical cell death pathway in the brain and eyes. *EMBO Mol Med* 5: 280-93
- Indrieri A, Grimaldi C, Zucchelli S, Tammaro R, Gustincich S, Franco B (2016) Synthetic long non-coding RNAs [SINEUPs] rescue defective gene expression in vivo. *Sci Rep* 6: 27315
- Indrieri A, van Rahden VA, Tiranti V, Morleo M, Iaconis D, Tammaro R, D'Amato I, Conte I, Maystadt I, Demuth S, Zvulunov A, Kutsche K, Zeviani M, Franco B (2012) Mutations in COX7B cause microphthalmia with linear skin lesions, an unconventional mitochondrial disease. *Am J Hum Genet* 91: 942-9
- Johnson SC, Yanos ME, Kayser EB, Quintana A, Sangesland M, Castanza A, Uhde L, Hui J, Wall VZ, Gagnidze A, Oh K, Wasko BM, Ramos FJ, Palmiter RD, Rabinovitch PS, Morgan PG, Sedensky MM, Kaerberlein M (2013) mTOR inhibition alleviates mitochondrial disease in a mouse model of Leigh syndrome. *Science* 342: 1524-8
- Khan NA, Nikkanen J, Yatsuga S, Jackson C, Wang L, Pradhan S, Kivela R, Pessia A, Velagapudi V, Suomalainen A (2017) mTORC1 Regulates Mitochondrial Integrated Stress Response and Mitochondrial Myopathy Progression. *Cell Metab* 26: 419-428 e5
- Kruse SE, Watt WC, Marcinek DJ, Kapur RP, Schenkman KA, Palmiter RD (2008) Mice with mitochondrial complex I deficiency develop a fatal encephalomyopathy. *Cell Metab* 7: 312-20
- Siegmund SE, Yang H, Sharma R, Javors M, Skinner O, Mootha V, Hirano M, Schon EA (2017) Low-dose rapamycin extends lifespan in a mouse model of mtDNA depletion syndrome. *Hum Mol Genet* 26: 4588-4605
- Song L, Yu A, Murray K, Cortopassi G (2017) Bipolar cell reduction precedes retinal ganglion neuron loss in a complex I knockout mouse model. *Brain Res* 1657: 232-244
- Yu AK, Song L, Murray KD, van der List D, Sun C, Shen Y, Xia Z, Cortopassi GA (2015) Mitochondrial complex I deficiency leads to inflammation and retinal ganglion cell death in the Ndufs4 mouse. *Hum Mol Genet* 24: 2848-60

3rd Editorial Decision

7 March 2019

Thank you for the submission of your revised manuscript to EMBO Molecular Medicine. We have now received the enclosed reports from the referees that were asked to re-assess it. As you will see the reviewers are now globally supportive and I am pleased to inform you that we will be able to accept your manuscript pending the following final amendments:

- 1) Please address the minor changes commented by the referees.
Please address both referees' comments in writing.

Please submit your revised manuscript within two weeks. I look forward to seeing a revised form of your manuscript as soon as possible.

I look forward to reading a new revised version of your manuscript as soon as possible.

***** Reviewer's comments *****

Referee #1 (Remarks for Author):

The authors have addressed all previous concern.

Please note that a typo is present in the figure representing the model (degradadion instead of degradation).

Referee #2 (Remarks for Author):

This is the fourth time the manuscript is submitted to EMM. They have made improvements to the work and the presentation. Overall, I would be favorable to accept it with the following caveats. The authors have elected not to present data on the whole *NDUFS4*^{-/-} mouse (i.e., survival). They say it is because the *n* is small, but the mouse only lives 50 days, so they would have had time to increase the *n*. I suspect that they want to keep the data for another manuscript, which is fine. However, they have to accept that they show data for the eye and the title and abstract must reflect this limitation. Second, the acute model with rotenone injected in the eye is inadequate and does not address or dissect any mechanism at all (my point 5 - rev 2); this figure has to go. The significance of the increase in complex I activity in *Ndufs4*^{-/-} *miR-181a/b*^{-/-} mice remains unclear (point 7) and the explanation (summarized in fig 7) somehow unconvincing. Again, this could be omitted or at least the interpretation should be tempered.

2nd Revision - authors' response

14 March 2019

Referee #1 (Remarks for Author):

The authors have addressed all previous concern.

Please note that a typo is present in the figure representing the model (degradadion instead of degradation).

We fixed this typo in Figure 7

Referee #2 (Remarks for Author):

This is the fourth time the manuscript is submitted to EMM. They have made improvements to the work and the presentation. Overall, I would be favorable to accept it with the following caveats. The authors have elected not to present data on the whole *NDUFS4*^{-/-} mouse (i.e., survival). They say it is because the *n* is small, but the mouse only lives 50 days, so they would have had time to increase the *n*. I suspect that they want to keep the data for another manuscript, which is fine. However, they have to accept that they show data for the eye and the title and abstract must reflect this limitation.

We modified the Abstract to more clearly state that the protective effect exerted by *miR-181a/b* downregulation was predominantly (although not exclusively, see MLS models) in the retina. The text now reads (Abstract, line 9: “We found that *miR-181a/b* downregulation strongly protects retinal neurons from cell death”).

Second, the acute model with rotenone injected in the eye is inadequate and does not address or dissect any mechanism at all (my point 5 - rev 2); this figure has to go.

We respectfully disagree with the Reviewer on this issue. We believe that the value of the rotenone data does not reside in the dissection of the molecular mechanisms but mostly in providing an independent evidence of the protective effect of *miR-181* down regulation across different mitochondrial disease models. Therefore, we feel that the rotenone data (and Figure 4) should not be removed from the manuscript. However, to better clarify the insights provided by the use of the rotenone model in the paper, we have added, in the revised text, the following statement: Results, Page 10, lines 16-19: “Rotenone-injected mice are considered a reliable model of LHON (Carelli et al, 2013). However, the latter represents an acute model in which it is difficult to dissect the mechanism by which *miR-181a/b* downregulation exert its protective function. Therefore, we decided to exploit the *Ndufs4*^{-/-} mouse model (Kruse et al, 2008) to...”

The significance of the increase in complex I activity in *Ndufs4*^{-/-} *miR-181a/b*^{-/-} mice remains unclear (point 7) and the explanation (summarized in fig 7) somehow unconvincing. Again, this could be omitted or at least the interpretation should be tempered.

To address the Reviewer's comment, we modified the text in the Discussion (page 15, lines 11-14), which now reads: "We hypothesize that this enhancement of complex activities is mediated by miR-181a/b action on degradation of the most dysfunctional mitochondria and enhanced mitochondrial biogenesis..."

Corresponding Author Name: Brunella Franco
Journal Submitted to: EMBO Molecular Medicine
Manuscript Number: EMM-2017-08734-V3